# Regulatory genome annotation of 33 insect species

**Hasiba Asma[1], Ellen Tieke[2], Kevin D Deem[2†], Jabale Rahmat[2], Tiffany Dong[3], Xinbo Huang[3], Yoshinori Tomoyasu[2], Marc S Halfon[1,3,4,5]***

[1]Program in Genetics, Genomics, and Bioinformatics, University at Buffalo-State University of New York, Buffalo, United States; [2]Department of Biology, Miami University, Oxford, United States; [3]Department of Biochemistry, University at Buffalo-State University of New York, Buffalo, United States; [4]Department of Biomedical Informatics, University at Buffalo-State University of New York, Buffalo, United States; [5]Department of Biological Sciences, University at Buffalo-State University of New York, Buffalo, United States

*For correspondence:
mshalfon@buffalo.edu

Present address: †Department of Biology, University of Rochester, Rochester, United States

Competing interest: The authors declare that no competing interests exist.

**Abstract** Annotation of newly sequenced genomes frequently includes genes, but rarely covers important non-coding genomic features such as the *cis*-regulatory modules—e.g., enhancers and silencers—that regulate gene expression. Here, we begin to remedy this situation by developing a workflow for rapid initial annotation of insect regulatory sequences, and provide a searchable database resource with enhancer predictions for 33 genomes. Using our previously developed SCRM-shaw computational enhancer prediction method, we predict over 2.8 million regulatory sequences along with the tissues where they are expected to be active, in a set of insect species ranging over 360 million years of evolution. Extensive analysis and validation of the data provides several lines of evidence suggesting that we achieve a high true-positive rate for enhancer prediction. One, we show that our predictions target specific loci, rather than random genomic locations. Two, we predict enhancers in orthologous loci across a diverged set of species to a significantly higher degree than random expectation would allow. Three, we demonstrate that our predictions are highly enriched for regions of accessible chromatin. Four, we achieve a validation rate in excess of 70% using in vivo reporter gene assays. As we continue to annotate both new tissues and new species, our regulatory annotation resource will provide a rich source of data for the research community and will have utility for both small-scale (single gene, single species) and large-scale (many genes, many species) studies of gene regulation. In particular, the ability to search for functionally related regulatory elements in orthologous loci should greatly facilitate studies of enhancer evolution even among distantly related species.

## eLife assessment

In the revised version of this **important** study, the authors present a **convincing** pipeline for insect genome regulatory annotation across 33 insect genomes spanning 5 orders. Despite technical limitations in the field owing to the lack of comprehensive knowledge of enhancer content in any system, the authors employ several independent downstream analyses to support the validity of their enhancer predictions for a subset of these genomes. Taken together, the revised results suggest that this prediction pipeline may have uses in identifying functional enhancers across large phylogenetic distances. Reviewers note caveats that an experimental validation is not yet available in the field to validate a large class of newly identified enhancers across such evolutionary distances, and other pipelines might be of use to compare. This work will be of interest to the computational genomics, evolutionary biology, and gene regulation fields.

## Introduction

The past two decades have witnessed an explosive rise in sequenced metazoan genomes, from a mere handful in the first few years of the century to over 8000 today (**NCBI, 2024**; accessed January 16, 2024). This impressive statistic, however, masks the reality that these genomes exist in various stages of completion. Fewer than 30% of these genomes are assembled at the chromosome level, and only 28% of those have a comprehensive annotation (**NCBI, 2024**). Moreover, almost none of the genome annotations include regulatory sequences (also referred to as *cis*-regulatory modules [CRMs]) such as enhancers and silencers. This is unfortunate, as CRMs comprise a significant percentage of the genome, and knowledge of these sequences is expected to have value comparable to that of knowing the protein-coding genes. Characterizing CRMs is critical for understanding mechanisms of gene regulation and the organization of gene regulatory networks. Moreover, the role of regulatory mutations is increasingly recognized as a driver of both evolution and disease (**Carroll et al., 2005**; **Carroll, 2008**; **Claringbould and Zaugg, 2021**; **Rickels and Shilatifard, 2018**; **Smith and Shilatifard, 2014**).

One reason for the overall dearth of regulatory annotations is that, historically, large-scale CRM discovery has been difficult and both resource and labor intensive (**Suryamohan and Halfon, 2015**). For much of the last four decades, CRMs could only be identified through painstaking, low-throughput experimental assays. Although in recent years high-throughput empirical and computational CRM discovery methods have been developed, the various different methods frequently show limited agreement (**Benton et al., 2019**; **Halfon, 2019**; **Lindhorst and Halfon, 2023**), with the result that comprehensive CRM annotation across all cell types and life-cycle stages has remained a challenge for all but the most exhaustively studied model organisms. The problem is particularly acute for the insects. Insects represent a species-rich class—they constitute somewhere between 65% and 90% of all animal species (**IUCN, 2022**; **Royal Entomological Society, 2023**)—and have major impacts on human health and agriculture. The early radiation of the insects, coupled with typically short generation times, means that most of the relevant biomedically and agriculturally important species share little non-coding sequence conservation with each other or with the principal insect model species, *Drosophila melanogaster*. Thus, common sequence-homology-based CRM discovery approaches are of little use in providing regulatory insights into these species, and knowledge transfers poorly from one species to another. Furthermore, many insects have a complex and varied life cycle, making it particularly important—yet onerous—to assay for CRM function at multiple stages.

We previously developed a powerful computational method, SCRMshaw ('Supervised Cis-Regulatory Module prediction'), for accurate prediction of CRMs, particularly enhancers (**Kantorovitz et al., 2009**; **Kazemian et al., 2011**; **Kazemian and Halfon, 2019**). (Although we expect SCRM-shaw to be adept at finding multiple CRM types, our validation efforts to date have focused solely on enhancers.) SCRMshaw requires only a sequenced genome and a 'training set' of some 15–30 known enhancers that regulate a common pattern of gene expression (e.g. midgut expression) and relies on the idea that enhancers with similar function will have similar sequence characteristics, not possible to detect by eye or by traditional alignment methods, but identifiable using machine learning. Although not universally true, this assumption is robust enough to allow effective enhancer discovery without requiring knowledge of transcription factor binding sites or of the expression patterns of the genes being regulated. Importantly, we have shown that we can leverage the wealth of existing *D. melanogaster* enhancer data (**Keränen et al., 2022**) to train models for *cross-species* supervised enhancer discovery in diverged (160–345 million years [MY]) insect species—including flies, mosquitoes, beetles, bees, and wasps—despite a virtually complete lack of observable alignment at the non-coding sequence level (**Kazemian et al., 2014**; **Lai et al., 2018**; **Schember and Halfon, 2021**; **Suryamohan et al., 2016**).

Here, we use SCRMshaw to undertake an initial regulatory annotation of 33 individual insect genomes, using a collection of 48 training sets composed of experimentally validated *D. melanogaster* enhancers. These species are spread across five orders spanning over 360 MY of evolution, and represent roughly 10% of annotated insect species with scaffold-level or better assembly. Annotated predicted enhancers are provided in a searchable database that allows querying by species, tissue/cell type, or potential target gene. A series of simulations as well as in silico and in vivo validation experiments demonstrate the effectiveness of our approach and place an upper bound on false-positive prediction rates. Our results represent the first release of a rich insect regulatory genome annotation

resource, which will continue to grow and annotate insect regulatory genomes in parallel with the sequencing of new insect genomes.

## Results

We previously demonstrated that SCRMshaw is remarkably effective at predicting enhancers across the entire ~345 MY range of the holometabolous insects, using training data derived solely from *D. melanogaster* (*Kazemian et al., 2014*; *Schember and Halfon, 2021*). SCRMshaw (*Figure 1A*) uses training sets composed of known enhancers defined by a common functional characterization (e.g. 'nervous system,' 'wing disc') to build a statistical model that captures their short DNA subsequence (*kmer*) count distribution. This *kmer* distribution is then compared to that of a set of non-enhancer 'background' sequences in a machine-learning framework. The *kmers* likely serve as proxies for the unknown transcription factor binding sites, but these sites themselves, even when known, are not explicitly used by the algorithm. The trained model is then used to score overlapping sequence windows in the genome, and the highest-scoring windows are output as predicted enhancers (*Kantorovitz et al., 2009*; *Kazemian et al., 2011*; *Asma and Halfon, 2019*). When searching the genomes of the mosquitoes *Anopheles gambiae* and *Aedes aegypti,* the red flour beetle *Tribolium castaneum,* the honey bee *Apis mellifera,* and the wasp *Nasonia vitripennis*, SCRMshaw successfully predicted enhancers in a cross-species fashion with an approximately 75% prediction success rate, based on reporter gene assays in xenotransgenic *D. melanogaster* (chosen as a pragmatic transgene host species) and comparison to already-identified enhancers in the other species (*Kazemian et al., 2014*; *Lai et al., 2018*; *Schember and Halfon, 2021*; *Suryamohan et al., 2016*). These results suggest that there are significant, albeit hidden, homologies governing the sequence characteristics of insect enhancers, at least for those involved in a substantial number of gene regulatory networks, and motivated us to apply SCRMshaw to a large and diverse set of sequenced insect genomes.

### A cross-species SCRMshaw pipeline

To facilitate application of SCRMshaw to large numbers of newly sequenced genomes, we developed a detailed workflow to ensure proper formatting of input genomes, rapid prediction of tissue-specific enhancer sequences, evaluation of results, and annotation of loci with information drawn from the respective orthologous regions in the richly investigated *D. melanogaster* genome (*Figure 1B*; see Methods). The workflow consists of four major steps:

#### Preflight

SCRMshaw requires two input files for each genome: a FASTA-formatted file of the genome sequence itself, and a GFFv3-formatted file of the genome annotation. The genome file is masked for tandem repeats using Tandem Repeat Finder (*Benson, 1999*). *Preflight* validates the formats of these files and produces a comprehensive log file that highlights any issues along with basic information such as the number of chromosomes/scaffolds and their sizes, data types present in the annotation (e.g. 'gene', 'exon', 'ncRNA', etc.), and average intergenic distances. *Preflight* also provides a sample output of the SCRMshaw-generated 'gene' and 'exon' files. This feature allows users to identify any discrepancies or errors stemming from the input files and to reformat these files as needed before running SCRMshaw. Any minor scaffolds that are annotated as not containing genes are discarded.

#### SCRMshaw

SCRMshaw is run as previously described (*Kazemian and Halfon, 2019*), using the 'HD' variant (*Asma and Halfon, 2019*). SCRMshaw_HD scans a genome with a 500 bp sliding window offset in 10 bp increments, using a set of three statistical models that compare the *kmer* composition of a set of training enhancers from *D. melanogaster* to randomly selected *D. melanogaster* non-coding sequences (see Methods).

#### Post-processing

The raw SCRMshaw output is post-processed to determine the final set of predicted enhancers. The original post-processing procedure described in *Asma and Halfon, 2019*, had a tendency to predict enhancers skewed toward long lengths (median ~1100 bp). We have revised that method here (see

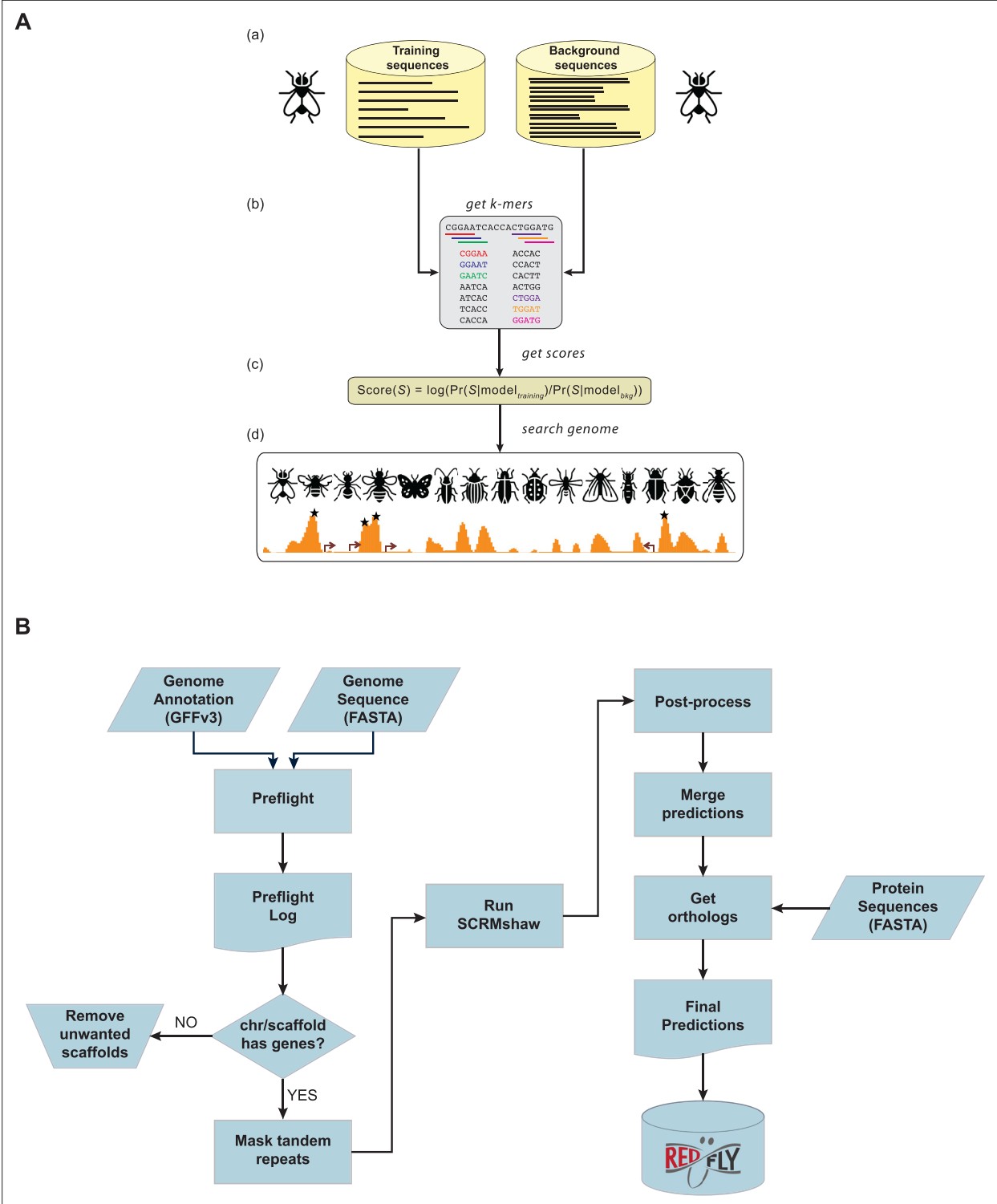

**Figure 1.** The SCRMshaw method and analysis pipeline. (**A**) Supervised motif-blind *cis*-regulatory module (CRM) discovery (SCRMshaw). (**a**) SCRMshaw uses a training set of known *D. melanogaster* enhancers ('training sequences'), drawn from REDfly, that are defined by common functional characterization, and a 10-fold larger background set of similarly sized common functional characterization, non-enhancer sequences ('background sequences'). (**b**) The short DNA subsequence (*kmer*) count distributions of these sequences are then used to train a statistical model. Note that although the pictured example shows 5-mers, *kmers* of different sizes are used for some of the underlying statistical models (see Methods). The trained model (**c**) is used to score overlapping windows in the 'target genome'. (**d**) High-scoring regions are predicted to be functional regulatory sequences (asterisks). Figure adapted from ***Asma and Halfon, 2021***. (**B**) The workflow used for the regulatory genome annotation described in this paper. The left side shows

*Figure 1 continued on next page*

*Figure 1 continued*

pre-processing steps, the right side, post-processing. Input to SCRMshaw consists of the genome sequence and gene annotation. A protein sequence annotation is supplied later for the orthology mapping step. Final results are made available as part of the REDfly regulatory annotation knowledgebase.

The online version of this article includes the following figure supplement(s) for figure 1:

**Figure supplement 1.** Revised post-processing method used for SCRMshaw.

Methods) to yield predictions of more compact size (median 750 bp), which is more in keeping with empirically characterized enhancer lengths (*Li et al., 2007*).

## Orthology mapping

Each SCRMshaw prediction is assigned its closest flanking genes as putative enhancer target genes, to aid in interpretation of the SCRMshaw output. There are clear shortcomings to this approach, as many enhancer targets are not the closest genes, leading to mis-assignments (e.g. *Sanyal et al., 2012*; *Hafez et al., 2017*; *Chua et al., 2022*; *Qin et al., 2022*). However, potentially more accurate methods for target-gene assignment rely on gene expression data, epigenetic data, and/or chromatin conformation data that are frequently not available for the species we are studying and thus difficult to incorporate into our prediction pipeline (*Qin et al., 2022*; *Fishilevich et al., 2017*; *Gschwind et al., 2023*; *Whalen et al., 2016*). Putative target genes are then mapped to their *D. melanogaster* orthologs (if an ortholog exists) using the *Orthologer* software from the Zdobnov lab (*Kuznetsov et al., 2023*) (see Methods). We use *D. melanogaster* as it has by far the most comprehensive gene annotation of the insects and thus provides the most detailed functional information for each gene. Mapping the genes from each species to a common ortholog helps us to assess whether we have obtained predicted enhancers in orthologous loci within the various species on which we have run SCRMshaw. The orthology mapping step requires that a set of predicted proteins is present as part of the existing genome annotation. Note that neither this step nor the earlier target-gene assignment step affects the enhancer predictions themselves, which are generated prior to target-gene assignment and orthology mapping and are considered equally valid regardless of putative target genes and whether or not orthologs can be matched to them.

## Annotation of 33 insect genomes

We ran our annotation pipeline on an initial set of 33 genomes (additional genome annotation is ongoing). These initial genomes were chosen based on availability and to sample broadly among the holometabolous insect orders and the Hemiptera (*Figure 2A*; *Table 1*). For each genome, we ran SCRMshaw using a collection of 48 training sets (*Figure 2—source data 1*) and all three SCRMshaw scoring methods ('IMM', 'hexMCD', 'PAC-rc'; *Kantorovitz et al., 2009*; *Kazemian et al., 2011*). For 15 species where a protein annotation was available, we assigned *D. melanogaster* orthologs to each predicted locus, with an average of 54% (range 38–82%) of genes in a given species having a *D. melanogaster* ortholog (*Figure 2B*). The complete data are available in multiple formats (see Data Availability).

Collectively, we predicted a total of 2,873,192 enhancers in these 33 species, with each species having on average approximately 87,000 predictions (*Figure 2—source data 2*; *Figure 2C*). (As some enhancers are predicted by multiple scoring methods, or from more than one training set, the number of unique sequences is lower at 1,164,354; see gray bars in *Figure 2C* and *Figure 2—source data 2*.) The median length of the predicted enhancers across all species was 750 bp (mean 695, range 490–32,500 bp). However, we noted the presence of a small number—2642, <0.1% of the total—of unusually large regions (>2000 bp), the bulk of which were confined to just a few genomes (*Figure 2—figure supplement 1*). We therefore discarded any predictions with length greater than 1.5 times the interquartile range of the complete prediction set and re-evaluated the size distribution. This resulted in a median size of 740 bp with a mean of 676 bp and a range of 490–1120 bp, indicating that the overall impact of these outlier sequences is minimal (*Figure 2D*). Inspection of the excessively large elements revealed that they result from SCRMshaw predictions that lie immediately adjacent to each other (without gaps) and have similar SCRMshaw scores, which thus become merged into one broad predicted element. Preliminary analysis suggests that these regions result from insufficient masking of tandem repeat regions and/or genome assembly errors, although other causes, such as extremely

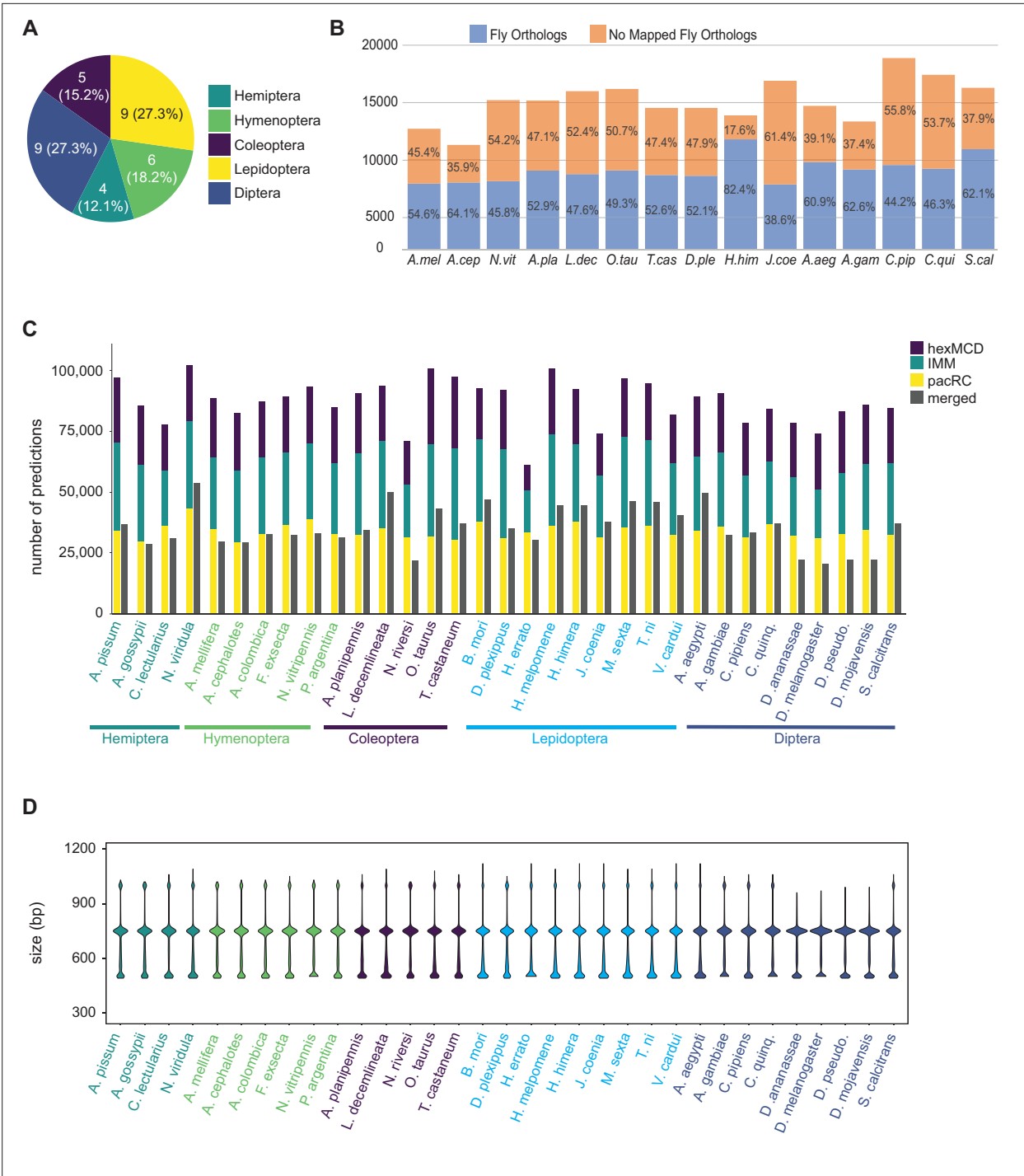

**Figure 2.** Annotation of 33 insect genomes. (**A**) Genomes from five insect orders were annotated in this study (more are ongoing). (**B**) Percentage of genes with *Drosophila* orthologs as mapped via our orthology pipeline (see Methods), for the 15 mapped species. 'No Mapped Fly Orthologs' indicates that our orthology mapping pipeline did not identify clear *D. melanogaster* orthologs. For any given gene, this could reflect either a true lack of a respective ortholog, or failure of our procedure to accurately identify an existing ortholog. For complete species names, see **Table 1**. (**C**) Total SCRMshaw predictions for each species. For each species, the left-hand column shows cumulative results for each SCRMshaw sub-method summed over each of the 48 training sets. The right-hand column shows the number of unique predictions after merging overlapping predictions from both sub-methods and training sets. Species are displayed alphabetically by taxonomic order (see also **Figure 2—figure supplement 1**). (**D**) Size distribution of SCRMshaw predictions, prior to merging overlapping predictions but after removing outlier predictions >2 kb in length. Species are ordered identically to panel C.

The online version of this article includes the following source data and figure supplement(s) for figure 2:

*Figure 2 continued on next page*

*Figure 2 continued*

**Source data 1.** SCRMshaw training sets used in this study.

**Source data 2.** Number of predicted enhancers for each species, by method.

**Figure supplement 1.** Lengths of predicted enhancers, including long outliers.

---

enhancer-dense 'superenhancer' regions (reviewed by *Grosveld et al., 2021*), cannot entirely be ruled out.

## Many loci contain multiple enhancer predictions

We have noted in the past that SCRMshaw often predicts multiple enhancers in a single locus (e.g. *Weinstein et al., 2023*). This is consistent with the concept of 'shadow enhancers', sets of multiple redundant or semi-redundant enhancers regulating the same gene (reviewed by *Kvon et al., 2021*). On the other hand, if SCRMshaw is predicting enhancers with low specificity (i.e. largely at random), it is instead possible that larger loci may just accumulate a high number of predictions simply due to their greater length.

To distinguish between these possibilities, we conducted simulations on three representative genomes (*D. melanogaster, C. pipiens,* and *A. aegypti*), each with a different average intergenic region size chosen to represent small, medium, and large genomes respectively (average sizes 610 bp, 5274.5 bp, and 14,641 bp; see Methods). We randomized the location of SCRMshaw predictions across the non-coding component of each genome and compared the randomized results to our SCRMshaw output. We found that, for a subset of loci, the number of real SCRMshaw predictions per locus was consistently higher than the number obtained at random (*Figure 3—source data 1*). For example, using the *mapping1.visceral_mesoderm* training set, 4.6% (15/328) of *D. melanogaster* loci containing one or more SCRMshaw predictions had a significantly (p<0.001) larger number of predictions/locus than expected, 1.5% (10/658) of *C. pipiens* loci had more predictions than expected, and 1% (3/299) of *A. aegypti* loci had excess predicted enhancers (*Figure 3—source data 1*). Similarly, for the *mapping2.ectoderm* training set, 7.6%, 1.5%, and 11% of loci in the three species, respectively, had a significantly (p<0.001) greater than expected number of predictions per locus (*Figure 3—source data 1*). Averaged over all training sets, 3–4% of loci with predictions had significantly more than expected by chance (mean values: 3.0% for *D. melanogaster*, 3.0% for *C. pipiens*, and 3.7% for *A. aegypti*; maximum values: 10.2% for *D. melanogaster*, 9.3% for *C. pipiens*, and 11.6% for *A. aegypti*). Only very few training sets did not have any loci with significantly more enhancers than expected (1 set in *A. aegypti*, 1 in *C. pipiens*, and 8 in *D. melanogaster*; *Figure 3—source data 1*).

To further ensure that these results were not influenced by locus size, we binned the loci by length and assessed the numbers of real versus simulated predictions/locus for each bin. The binned results were similar to the results using all loci, i.e., the number of SCRMshaw predictions/locus (*Figure 3*, blue boxplots) was consistently higher than the number obtained via randomization (*Figure 3*, pink boxplots). Results were also similar when comparing predictions only at intergenic versus only at intronic positions (*Figure 3—source data 1*). These results suggest that SCRMshaw is not predicting enhancers randomly throughout the genome, but rather is identifying multiple related 'shadow' enhancers in a subset of loci. Consistent with this interpretation, functional tests of the full set of predicted enhancers in several *D. melanogaster* loci has confirmed that many of the predictions act as functionally similar enhancers (T Williams, personal communication, December 2023).

## SCRMshaw predicts enhancers in orthologous loci across species

A major premise underlying cross-species applications of SCRMshaw is that conserved regulatory strategies should allow for a model trained on enhancers in one species to predict for similar enhancers in another species. We reasoned that at least some of the time, this should lead to identification of enhancers in orthologous loci, as orthologous genes are frequently involved in similar biological pathways and developmental regulatory networks (*Carroll, 2008*). Indeed, we previously showed that SCRMshaw was able to predict enhancers in several orthologous loci, for instance those for the *single-minded* locus in *D. melanogaster, A. gambiae,* and *A. mellifera,* and the *wingless* locus in *D. melanogaster, A. mellifera,* and *T. castaneum* (*Kazemian et al., 2014*). To test whether this is generally true, we used the 15 species with mapped *D. melanogaster* orthologs (plus *D. melanogaster* itself), and

**Table 1.** Species used in this study.

| Scientific name | Common name | Order | Assembly version/ annotation version | Link/URL for assembly information |
|---|---|---|---|---|
| *Acyrthosiphon pisum* | Pea aphid | Hemiptera | racon and v3_wdel | Courtesy Jennifer Brisson (University of Rochester) |
| *Aedes aegypti* | Yellow fever mosquito | Diptera | L5.2 | https://www.ncbi.nlm.nih.gov/datasets/taxonomy/7159/ |
| *Agrilus planipennis* | Emerald ash borer | Coleoptera | Apla_2.0 | https://www.ncbi.nlm.nih.gov/datasets/taxonomy/224129/ |
| *Anopheles gambiae* | African malaria mosquito | Diptera | P4.9 | https://www.ncbi.nlm.nih.gov/datasets/taxonomy/7165/ |
| *Aphis gossypii* | Cotton aphid/ melon aphid | Hemiptera | ASM401081v2 | https://www.ncbi.nlm.nih.gov/datasets/taxonomy/80765/ |
| *Apis mellifera* | Honey bee | Hymenoptera | HAv3.1 | https://www.ncbi.nlm.nih.gov/datasets/taxonomy/7460/ |
| *Atta cephalotes* | Leafcutter ant | Hymenoptera | A.ceph_1.0 | https://www.ncbi.nlm.nih.gov/datasets/taxonomy/12957/ |
| *Atta colombica* | Leafcutter ant | Hymenoptera | Acol1.0 | https://www.ncbi.nlm.nih.gov/datasets/taxonomy/520822/ |
| *Bombyx mori* | Silkworm | Lepidoptera | ASM15162v1 | http://ensembl.lepbase.org/Bombyx_mori_asm15162v1/Info/Index |
| *Cimex lectularius* | Bed bug | Hemiptera | Clec_2.1 | https://www.ncbi.nlm.nih.gov/datasets/taxonomy/79782/ |
| *Culex pipiens pallens* | Northern house mosquito | Diptera | TS_Cpip_V1 | https://www.ncbi.nlm.nih.gov/datasets/taxonomy/42434/ |
| *Culex quinquefasciatus* | Southern house mosquito | Diptera | VIPSU_Cqui_1.0_pri_paternal | https://doi.org/10.1093/gbe/evab005 |
| *Danaus plexippus* | Monarch butterfly | Lepidoptera | v3 | http://ensembl.lepbase.org/Danaus_plexippus_v3/Info/Index |
| *Drosophila ananassae* | Fruit fly | Diptera | caf1 | http://ftp.flybase.net/genomes/Drosophila_ananassae/ |
| *Drosophila melanogaster* | Fruit fly | Diptera | r6 1.8 | https://www.ncbi.nlm.nih.gov/datasets/taxonomy/7227/ |
| *Drosophila mojavensis* | Fruit fly | Diptera | caf1 | https://www.ncbi.nlm.nih.gov/assembly/GCF_000005175.2/ |
| *Drosophila pseudoobscura* | Fruit fly | Diptera | r3 | http://ftp.flybase.net/genomes/Drosophila_pseudoobscura/dpse_r3.03_FB2015_01/ |
| *Formica exsecta* | Wood ant | Hymenoptera | ASM365146v1 | https://www.ncbi.nlm.nih.gov/datasets/genome/GCF_003651465.1/ |
| *Heliconius erato* | Red postman butterfly | Lepidoptera | v1 | http://ensembl.lepbase.org/Heliconius_erato_lativitta_v1/Info/Index |
| *Heliconius melpomene* | Postman butterfly | Lepidoptera | Hmel2 | http://ensembl.lepbase.org/Heliconius_melpomene_melpomene_hmel2/Info/Index |
| *Heliconius himera* | False postman butterfly | Lepidoptera | Hed.V1 | Courtesy Robert Reed (Cornell University) |
| *Junonia coenia* | Common buckeye butterfly | Lepidoptera | JC v1.0 | Courtesy Robert Reed (Cornell University) |
| *Leptinotarsa decemlineata* | Colorado potato beetle | Coleoptera | Ldec_2.0 | https://www.ncbi.nlm.nih.gov/datasets/taxonomy/7539/ |
| *Manduca sexta* | Tobacco hornworm | Lepidoptera | v1.0 | http://ensembl.lepbase.org/Manduca_sexta_msex1/Info/Index |

*Table 1 continued on next page*

*Table 1 continued*

| Scientific name | Common name | Order | Assembly version/ annotation version | Link/URL for assembly information |
|---|---|---|---|---|
| *Nezara viridula* | Southern green stink bug | Hemiptera | | https://www.ncbi.nlm.nih.gov/datasets/taxonomy/85310/ |
| *Nasonia vitripennis* | Jewel wasp | Hymenoptera | Psr_1.1 | https://www.ncbi.nlm.nih.gov/datasets/taxonomy/7425/ |
| *Onthophagus taurus* | Dung beetle | Coleoptera | Otau_2.0 | https://www.ncbi.nlm.nih.gov/datasets/taxonomy/166361/ |
| *Pseudoatta argentina* | Leafcutter ant | Hymenoptera | ASM1760752v1 | https://www.ncbi.nlm.nih.gov/datasets/taxonomy/621737/ |
| *Stomoxys calcitrans* | Stable fly | Diptera | Stomoxys_calcitrans-1.0.1 | https://doi.org/10.1186/s12915-021-00975-9 |
| *Tribolium castaneum* | Red flour beetle | Coleoptera | r5.2 | https://www.ncbi.nlm.nih.gov/datasets/taxonomy/7070/ |
| *Trichoplusia ni* | Cabbage looper | Lepidoptera | tn1 | https://www.ncbi.nlm.nih.gov/datasets/taxonomy/7111/ |
| *Vanessa cardui* | Painted lady butterfly | Lepidoptera | Vcar_v1 | Courtesy Robert Reed (Cornell University) |
| *Nebria riversi* | Ground beetle | Coleoptera | v1 | https://doi.org/10.1111/1755-0998.13409 |

all 48 of our training sets, to compare the number of SCRMshaw-based versus random predictions obtained in common (orthologous) loci.

We observed a substantial reduction in the number of orthologous loci with predictions in common as we moved from considering a set of 10 to the full set of all 16 species (average number of common loci over all 48 training sets: 70.7 (10 species)>39.4>19.8>9.14>4.12>1.6>0.6 (16 species) out of an average number of 640 predictions with mapped orthologs) (*Figure 4A*, *Figure 4—source data 1*, tab 'a'). This rapid decline is likely due to a variety of factors, including differences in taxonomic order (e.g. Diptera vs. Hymenoptera), the quality and degree of ortholog data for each species, and the quality of annotation for each species, all of which likely lead to an underestimation of the true numbers of common loci in our data (see Discussion). We simulated SCRMshaw predictions for all 16 species by randomizing the SCRMshaw results and compared the number of common loci between the simulated and real data for combinations of 10–16 species. The number of common loci among the real SCRMshaw predictions was consistently significantly higher than the number of common loci observed in the simulated data ($p<0.05$; *Figure 4B* and *Figure 4—source data 1*). In particular, when evaluating between 10 and 12 species (data for 14–16 species are not reliable due to the very small number of observed loci in common) only a single training set, *adult_PNS*, did not have a significant overrepresentation of common loci (*Figure 4—source data 1*, tab 'c'). We also examined the fold enrichment, i.e., the extent to which the true number was in excess of the simulated number, and found that on average there were greater than 2.4× more predictions in common loci than expected, when considering groups of 10–12 species; almost all training sets had a fold enrichment >1.5× (*Figure 4C*, *Figure 4—source data 1*, tab 'd'). These results strongly suggest that SCRMshaw is successfully finding sets of conserved (or convergent) enhancers, as it is predicting sequences in orthologous loci in a training-set-specific manner across multiple species significantly more often than can be accounted for by chance.

## SCRMshaw predictions correlate with regions of accessible chromatin

Active enhancers are frequently found in regions of accessible chromatin, as assayed by methods such as DNAse-seq, FAIRE-seq (Formaldehyde-Assisted Isolation of Regulatory Elements with sequencing), and ATAC-seq (Assay for Transposase-Accessible Chromatin by sequencing) (*Boyle et al., 2008*; *Buenrostro et al., 2013*; *Giresi et al., 2007*). We confirmed previously that FAIRE-predicted and SCRMshaw-predicted enhancers in *T. castaneum* have a high (>79%) degree of overlap (*Lai et al., 2018*). To determine whether a similar correlation exists for other species, we compared our SCRMshaw predictions with available FAIRE-seq and ATAC-seq data for seven species: *D. melanogaster,*

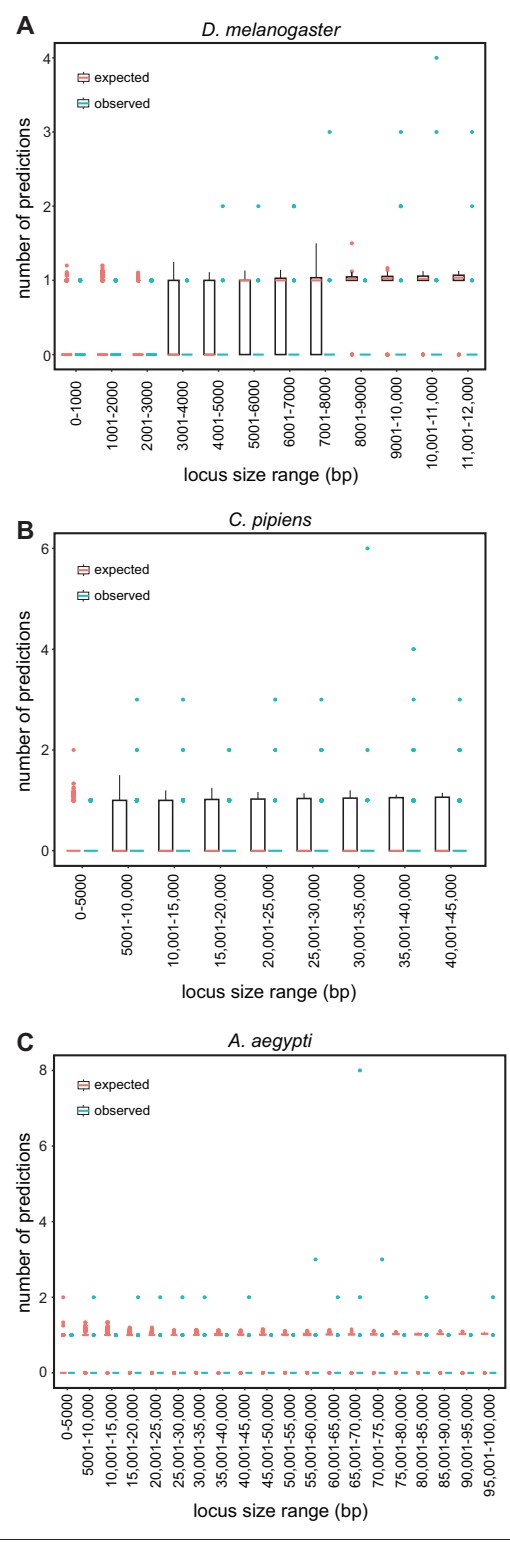

**Figure 3.** SCRMshaw makes multiple predictions per locus. The number of SCRMshaw predictions per locus (y-axis) are shown as boxplots for loci falling within the given size ranges (x-axis). Black boxes cover the 25–75th percentiles, bars indicate median values and dots indicate values exceeding 1.5 times the interquartile

*Figure 3 continued on next page*

*Figure 3 continued*

range (boxes are not visible for all bins due to very low degrees of variation). Values in pink represent expected values drawn from randomization, while values in blue represent observed values from SCRMshaw. All values are from results with the training set 'mapping1. visceral_mesoderm'; results from other training sets were similar (see *Figure 3—source data 1*). Shown are results from the genomes of (**A**) *D. melanogaster*, (**B**) *C. pipiens*, and (**C**) *A. aegypti* representing small, medium, and large genomes, respectively.

The online version of this article includes the following source data for figure 3:

**Source data 1.** Real and simulated predictions per locus.

---

*T. castaneum, A. gambiae, Danaus plexippus, Vanessa cardui, Junonia coenia,* and *Heliconius himera*. These comparisons are imperfect, as the tissues used to obtain the chromatin data do not precisely correspond to the training sequences used for SCRMshaw, and the data were obtained using a variety of methods. Nevertheless, in the majority of cases where we were able to establish a rough match between the tissues, we observed significant overlap between the two methods of enhancer detection. For example, in *D. melanogaster*, 40% of SCRMshaw predictions using the *blastoderm.mapping1* training set overlapped ATAC-seq peaks obtained from blastoderm embryos (p<4.15e-112, fold enrichment 2.98) (*Bozek et al., 2019*; *Table 2*, row 2). Similarly, 35% and 41% of SCRMshaw predictions from the *mappng2.wing* and *haltere_disc* sets overlapped FAIRE data that included wing and haltere cells (*McKay and Lieb, 2013*) (p<4.6e-117 and 1.5e-102, fold enrichment of 3.29 and 2.98) (*Table 2*, rows 1, 3). In the mosquito *A. gambiae*, we compared SCRMshaw predictions from the *embryonic_midgut* and *mapping1.salivary* training sets to ATAC-seq data from adult midgut and salivary tissues, observing overlaps of 40% and 37% respectively (p<1.5e-102 and 2.15e-93, fold enrichment of 3.45 and 3.22) (*Table 2*, rows 9, 10). When comparing our SCRMshaw predictions from the *mapping2.wing* set to ATAC-seq data for larval wing tissue in the butterflies *D. plexippus* and *H. himera*, we observed overlaps of 60% and 68% (p<5.59e-19 and p<9.92e-25, fold enrichment of 1.99, 1.66) (*Table 2*, rows 11, 12) (*Mazo-Vargas et al., 2022*). The butterflies *J. coenia* and *V. cardui* are exceptions; intriguingly, they show a depletion in SCRMshaw predictions compared to expectation (*Table 2*, rows 13, 14). Whether this is due to a mismatch in the data

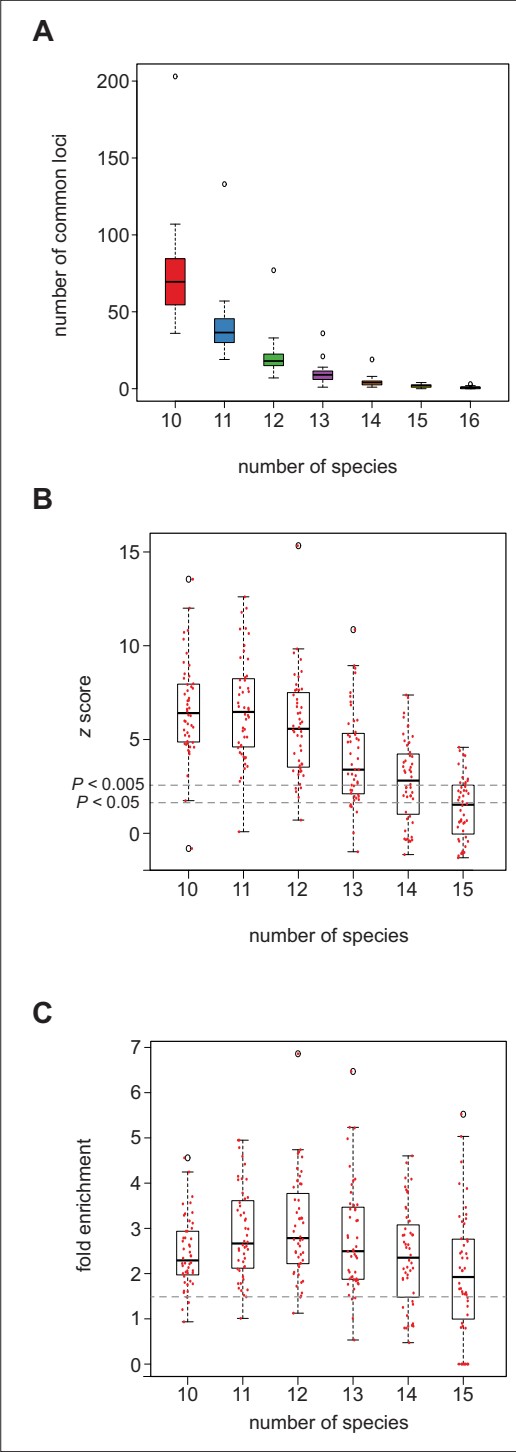

**Figure 4.** SCRMshaw predicts *cis*-regulatory modules (CRMs) in orthologous loci across species. (**A**) The number of loci in common that contain at least one SCRMshaw prediction, for 10 or more species. (**B**) z-scores demonstrating that the number of loci in common with one or more SCRMshaw predictions is significantly higher than expectation, based on 360 randomizations. The small number of common predictions for 14–16 species make these

*Figure 4 continued on next page*

*Figure 4 continued*

statistics unreliable. Dotted lines indicate z-score values representing significance at the (unadjusted) p<0.005 and p<0.05 levels. (**C**) Fold enrichment values illustrating the excess of loci in common with one or more SCRMshaw predictions compared to expectation. Dotted line shows 1.5× enrichment.

The online version of this article includes the following source data for figure 4:

**Source data 1.** Real and simulated counts of predictions in orthologous loci.

used for comparison, the state of the genome assemblies for these two draft genomes, or some other failure of SCRMshaw to perform strongly on these species remains to be determined. Overall, the highly significant overlap we observe between SCRMshaw predictions and open chromatin regions in most species and tissues provides further strong evidence that SCRMshaw is effectively predicting enhancers across a broad range of genomes.

## Reporter gene analysis demonstrates that SCRMshaw predictions are functional enhancers

As a concrete test of our ability to use SCRMshaw to predict functional enhancers, we assayed a subset of SCRMshaw predictions by reporter gene analysis in transgenic *D. melanogaster*. Our previous studies have shown a high success rate for such assays, ranging from 65% through >80%, depending on the species and training sets used (*Kantorovitz et al., 2009*; *Kazemian et al., 2011*; *Kazemian et al., 2014*; *Schember and Halfon, 2021*; *Suryamohan et al., 2016*).

We focused our in vivo validation experiments on SCRMshaw predictions made using the *mapping2.wing*, *haltere_disc*, and *disc.mapping2* training sets and a set of four species we had previously shown to be amenable to SCRMshaw prediction: *D. melanogaster*, *A. aegypti*, *T. castaneum*, and *A. mellifera*. The imaginal discs are well-established tissues for investigating gene regulation in *D. melanogaster* (although see Discussion for a caveat on using imaginal discs for evaluating enhancer activities in a cross-species setting), and the chosen training sets all gave significant results in the open-chromatin comparisons discussed above. We selected six sets of putative enhancers for testing (*Table 3*, *Table 4*). For the first three sets, we selected sequences where we had predictions in each of

**Table 2.** Overlap of SCRMshaw predictions with FAIRE-seq and ATAC-seq peaks.

| Species | Training set | Profiled tissue | Overlap (real) | Overlap (random) | s.d.* | z-score | FE[†] |
|---|---|---|---|---|---|---|---|
| D. melanogaster | haltere_disc | Wing, leg, and haltere third instar discs and pharate appendages; eye-antennal disc; third instar CNS | 41.82% | 14.02% | 10.65 | 21.53 | 2.98 |
| D. melanogaster | blastoderm.mapping1 | Blastoderm | 40.99% | 13.86% | 12.35 | 22.56 | 2.96 |
| D. melanogaster | mapping2.wing | Wing, leg, and haltere third instar discs and pharate appendages; eye-antennal disc; third instar CNS | 35.14% | 10.69% | 10.19 | 23.01 | 3.29 |
| T. castaneum | mapping2.wing | Embryo, larval thoracic epidermis, larval brain | 85.58% | 26.18% | 10.98 | 28.88 | 3.27 |
| T. castaneum | mapping2.ectoderm | Embryo, larval thoracic epidermis, larval brain | 81.68% | 25.48% | 12.48 | 26.29 | 3.21 |
| T. castaneum | mapping1.ventral_ectoderm | Embryo, larval thoracic epidermis, larval brain | 81.17% | 24.76% | 12.56 | 33.87 | 3.28 |
| T. castaneum | mapping1.dorsal_ectoderm | Embryo, larval thoracic epidermis, larval brain | 71.15% | 24.69% | 12.64 | 24.07 | 2.88 |
| T. castaneum | mapping1.ectoderm | Embryo, larval thoracic epidermis, larval brain | 69.53% | 24.66% | 11.92 | 22.35 | 2.82 |
| A. gambiae | embryonic_midgut | Adult midgut, salivary_gland | 40.20% | 11.66% | 7.90 | 21.56 | 3.45 |
| A. gambiae | mapping1.salivary_gland | Adult midgut, salivary_gland | 36.76% | 11.43% | 8.52 | 20.55 | 3.22 |
| D. plexippus | mapping2.wing | Larval forewing, hindwing, head | 60.43% | 30.35% | 6.30 | 8.93 | 1.99 |
| H. himera | mapping2.wing | Larval forewing, hindwing, head | 68.34% | 41.10% | 8.97 | 10.27 | 1.66 |
| J. coenia | mapping2.wing | Larval forewing, hindwing, head | 3.06% | 34.91% | 8.52 | −12.22 | 0.09 |
| V. cardui | mapping2.wing | Larval forewing, hindwing, head | 15.46% | 36.01% | 8.01 | −7.13 | 0.43 |

*Standard deviation.

[†]FE, fold enrichment.

the orthologous loci for *D. melanogaster, T. castaneum,* and *A. mellifera* (*A. aegypti* was not considered for these sets), and where the *D. melanogaster* prediction mapped near a gene expressed in the wing imaginal discs (*Figure 5*). These predictions were conducted using SCRMshaw's IMM scoring method only, with post-processing performed using the original method described in *Asma and Halfon, 2019*, and the Amel_4.5 version of the *A. mellifera* genome. For the second three sets, we chose sequences where we had predictions in orthologous loci for at least three of the four species and where the *D. melanogaster* sequence had previously been tested in a reporter gene assay and was known to be active in the relevant imaginal discs (*Figure 5*). Imposing this latter criterion allowed us to leverage existing knowledge and reduce the necessary amount of in vivo testing for each set of predictions, enabling us to test a larger set of sequences overall. For this second set of three, we used all three SCRMshaw scoring methods with a revised post-processing algorithm (see Methods), and the Amel_Hav3.1 *A. mellifera* genome. For all six sets, predictions were chosen for testing based on

**Table 3.** Gene loci chosen for in vivo validation.

| D. melanogaster | | | T. castaneum | | A. mellifera | A. aegypti |
|---|---|---|---|---|---|---|
| Name | Symbol | FlyBaseID | RefSeq | iBeetleBase | RefSeq | VectorBase |
| expanded | ex | FBgn0004583 | gene-LOC657053 | TC012545 | gene-LOC551519 | AAEL001437 |
| u-shaped | ush | FBgn0003963 | gene-LOC659918 | TC013689 | gene-LOC100577801 | AAEL020615 |
| klumpfuss | klu | FBgn0013469 | gene-LOC103312803 | TC002783 | gene-LOC100577692 | AAEL013544 |
| homothorax | hth | FBgn0001235 | gene-Hth | TC008629 | gene-LOC552079 | AAEL011643 |
| pipsqueak | psq | FBgn0263102 | gene-LOC660343 | TC003349 | gene-psq | AAEL021255 |
| Ultrabithorax | Ubx | FBgn0003944 | gene-Ubx | TC000903 | gene-ubx | AAEL014032 |

**Table 4.** SCRMshaw predictions chosen for in vivo validation.

| | Coordinates | Max. score | Training set(s) | Method |
|---|---|---|---|---|
| **Set 1** | | | | |
| *T. castaneum* | | | | |
| Tc_ex_9p0 | NC_007424.3:11221530..11222170 | 9.04 | mapping2.wing | imm |
| Tc_ush_6p8 | NC_007420.3:5968840..5969370 | 6.84 | haltere_disc | imm |
| Tc_klu_8p6 | NC_007418.3:10416700..10417300 | 8.59 | mapping2.wing | imm |
| *D. melanogaster* | | | | |
| Dm_ex_20p0 | 2L:442110..442810 | 20.04 | mapping2.wing | imm |
| Dm_klu_16p1 | 3L:10991040..10991700 | 16.06 | mapping2.wing | imm |
| Dm_ush_16p4 | 2L:531250..532060 | 16.40 | mapping2.wing | imm |
| *A. mellifera* | | | | |
| Am_ex_20p3 | NC_007075.3:7545750..7546440 | 20.36 | mapping2.wing | imm |
| Am_klu_20p2 | NC_007070.3:720220..720870 | 20.25 | mapping2.wing | imm |
| Am_ush_20p8 | NC_007080.3:10609550..10610200 | 20.80 | haltere_disc | imm |
| **Set 2** | | | | |
| *T. castaneum* | | | | |
| Tc_hth_15p5 | NC_007422.5:13408990..13409850 | 15.52 | mapping2.wing | hexmcd,imm,pac |
| Tc_psq_19p7 | NC_007418.3:1805000..1806250 | 19.71 | haltere_disc, mapping2.wing | hexmcd,imm,pac |
| Tc_Ubx_19p9 | NC_007417.3:8137250..8138000 | 19.32 | haltere_disc | hexmcd |
| Tc_Ubx_17p4 | NC_007417.3:8153250..8154030 | 19.95 | mapping2.wing | hexmcd,imm |
| *A. aegypti* | | | | |
| Aa_hth_35p9 | 1:149733960..149734710 | 35.96 | mapping2.wing | imm |
| Aa_psq_21p5 | 2:228190750..228191640 | 21.50 | disc.mapping2 | hexmcd,imm |
| Aa_Ubx_26p0 | 1:309747490..309748390 | 26.08 | mapping2.wing | imm |
| *A. mellifera* | | | | |
| Am_psq_29p2 | NC_007078.3:10712000..10712750 | 29.26 | mapping2.wing | hexmcd |
| Am_Ubx_37p2 | NC_007085.3:2921250..2922250 | 37.18 | haltere_disc, mapping2.wing | hexmcd,imm |
| Am_Ubx_0p39 | NC_007085.3:2967750..2968250 | 0.39 | mapping2.wing | pac |

high SCRMshaw scores and overlap with open-chromatin data (where available). We also considered the position of each prediction within the locus (i.e. first intron, downstream intergenic region, etc.), favoring sequences where position was maintained among the orthologs. Selected sequences were cloned into a cross-species compatible reporter vector (*Lai et al., 2018*; *Deem et al., 2024*), and reporter gene activity was visualized either directly or by using the lineage tracing system G-TRACE (*Evans et al., 2009*). In the latter, enhancer activity is visualized through two reporters; the first reporter visualizes the direct enhancer activity while expression of the second reporter is induced and maintained in all cells that descend from a cell in which the enhancer is initially active, even if activity subsequently shuts off.

## ex

The *expanded (ex)* gene plays a crucial role in tissue growth control, including wings (*Boedigheimer and Laughon, 1993*; *Wang and Baker, 2015*; *Wang and Baker, 2018*). There was a single prediction in the *D. melanogaster ex* locus, *Dm_ex_20p0*, which falls within an open chromatin region of the third intron (*Figure 6—figure supplement 1*), and which comprises an untested subsequence of a longer

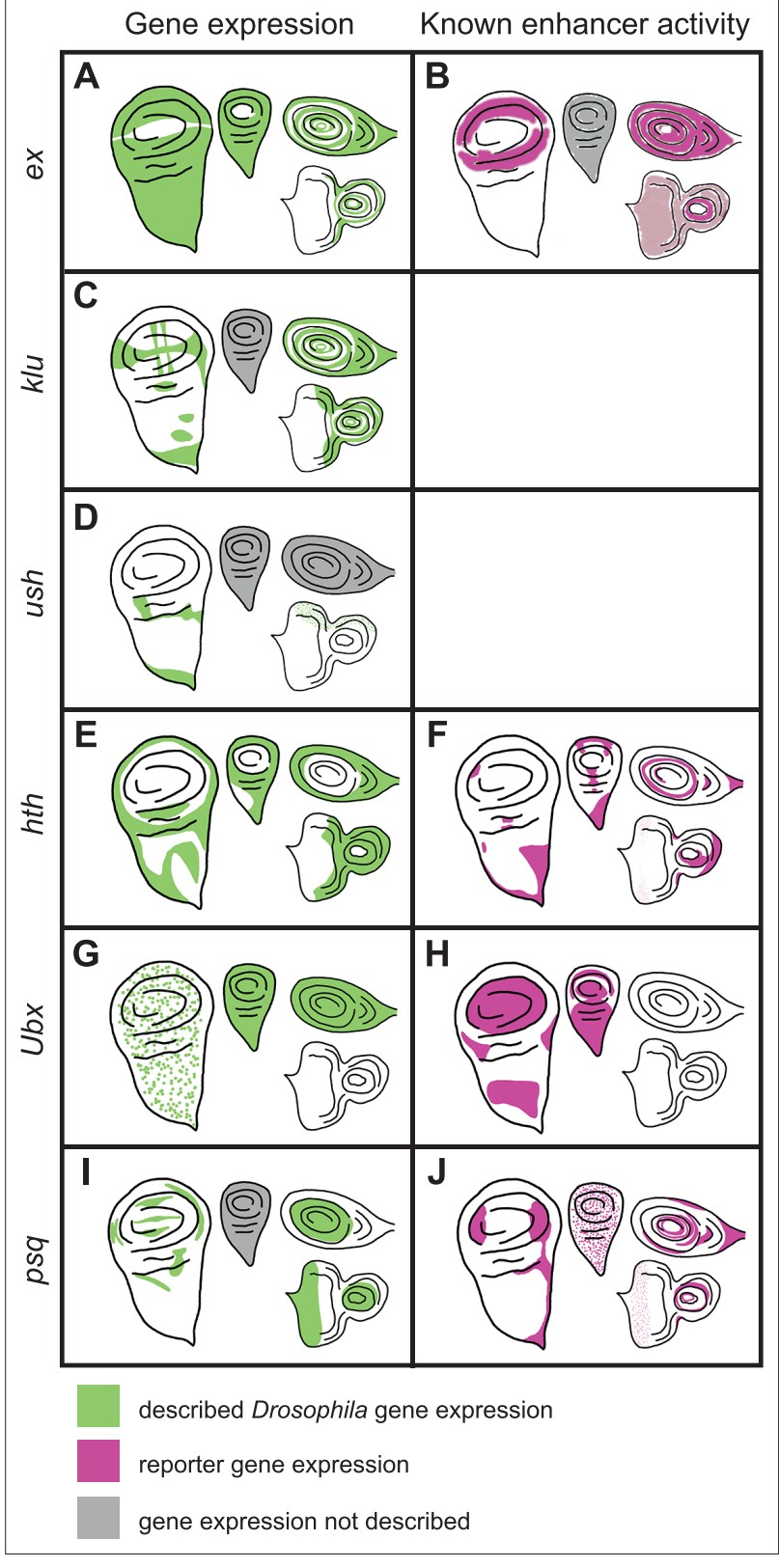

**Figure 5.** Previously described gene expression and enhancer activity for select *D. melanogaster* sequences predicted by SCRMshaw. The left-hand column shows native *D. melanogaster* gene expression in imaginal discs (green), while the right-hand column shows described enhancer activity (magenta). Gray shading indicates that expression has not been described. Moving clockwise from the left side of each panel are the wing, haltere, leg,

*Figure 5 continued on next page*

*Figure 5 continued*

and eye-antennal discs. The enhancers whose activities are described in the table are: (B) ex_BCDE (*Wang and Baker, 2018*), (F) hth_GMR46D04 (*Jory et al., 2012*), (H) Ubx_GMR39A02 (*Jory et al., 2012*), (J) psq_GMR41E12 (*Jory et al., 2012*).

---

sequence that acts as an imaginal disc enhancer (*Figure 5B*; *Wang and Baker, 2018*). This sequence drove reporter activity in the wing, leg, and antennal discs (*Figure 6A*). The *T. castaneum* genome also had only one prediction for the *ex* locus, *Tc_ex_9p0*, which overlaps well with a FAIRE-seq peak (*Figure 6—figure supplement 1*) within the second intron. Although no imaginal disc activity was observed (*Figure 6B*), *Tc_ex_9p0*-driven reporter gene expression was observed during late pupal stages in the legs (*Figure 6—figure supplement 2G*). The *A. mellifera* genome (v4.5) had one prediction, *Am_ex_20p3*, within the fourth intron of the *ex* locus (*Figure 6—figure supplement 1*; however, note that subsequent prediction using the updated Amel_Hav3.1 genome and revised SCRMshaw post-processing yielded additional predictions in this locus). *Am_ex_20p3* drove active but variable expression in both the pouch and notum regions of the wing imaginal disc (*Figure 6C*). Although often significantly limited to a small number of cells, the pouch expression of *Am_ex_20p3* was similar in pattern to that seen with *Dm_ex_20p0* (*Figure 6A*). Weak and inconsistent expression in the pouch region of the haltere discs was also observed. In addition, *Am_ex_20p3* drove expression in the leg discs (*Figure 6C*).

## klu

*Klumpfuss* (*klu*) encodes a zinc finger protein important for proper tissue specification and differentiation (*Klein and Campos-Ortega, 1997*). The *D. melanogaster* genome had three predictions for the *klu* locus (*Figure 6—figure supplement 1*). We selected *Dm_klu_16p1*, which overlaps a region of open chromatin within the second intron present in most imaginal discs (most distinct in the T3 leg disc; *Figure 6—figure supplement 1*). *Dm_klu_16p1* drove active expression in the notum regions of the wing and haltere discs and in the leg discs (*Figure 6D*). *Tc_klu_8p6* was the only prediction for the *T. castaneum klu* locus. It falls within the second intron, similar to *D. melanogaster's klu* prediction *Dm_klu_161p1* (*Figure 6—figure supplement 1*), but lacked enhancer activity (*Figure 6E*). *A. mellifera* had only one prediction for the *klu* locus (*Am_klu_20p2*) (*Figure 6—figure supplement 1*). We included this prediction for validation even though its location in the third intron does not match with that in the other two species. *Am_klu_20p2* drove weak expression in the most proximal part of the leg discs (*Figure 6F*, arrows), but no activity was observed in the other imaginal discs. (Note that additional predictions were subsequently produced with our revised post-processing algorithm and the updated Amel_Hav3.1 genome, for both the *T. castaneum* and *A. mellifera klu* loci.)

## ush

Our final choice from our first set for in vivo validation was *u-shaped* (*ush*), which encodes a transcription factor with described imaginal disc expression (*Buchberger et al., 2021*; *Cubadda et al., 1997*; *Tomoyasu et al., 2000*). The *D. melanogaster* genome had two predictions for the *ush* locus (three when using the updated post-processing algorithm) (*Figure 6—figure supplement 1*). We tested *Dm_ush_16p4*, but did not observe any larval disc activity (*Figure 6G*). *Tc_ush_6p8* and *Am_ush_20p8* were the only predictions for *T. castaneum* and *A. mellifera*, respectively, both located in the second intron (again, additional predictions are found using updated methods and genome versions). *Tc_ush_6p8* had expression in the wing disc as well as weak activity in the peripodial membrane of the eye-antennal disc (*Figure 6H*, *Figure 6—figure supplement 2K*). *Am_ush_20p8* displayed active expression in the peripodial membrane surrounding the eye disc, as well as in a single cell, or small subset of cells, located at the center of each leg disc (*Figure 6I*, arrows).

## hth

SCRMshaw predicted 10 enhancers in the locus of the *D. melanogaster* Hox cofactor *homothorax* (*hth*). One of the two top-scoring predictions, *Dm_hth_30p1*, located in the fifth intron, overlapped the known *hth_GMR46D04* enhancer, which has activity in all of the larval imaginal discs (*Figure 5F*, *Figure 7—figure supplement 1*; *Jory et al., 2012*). *A. aegypti* had a single prediction in the

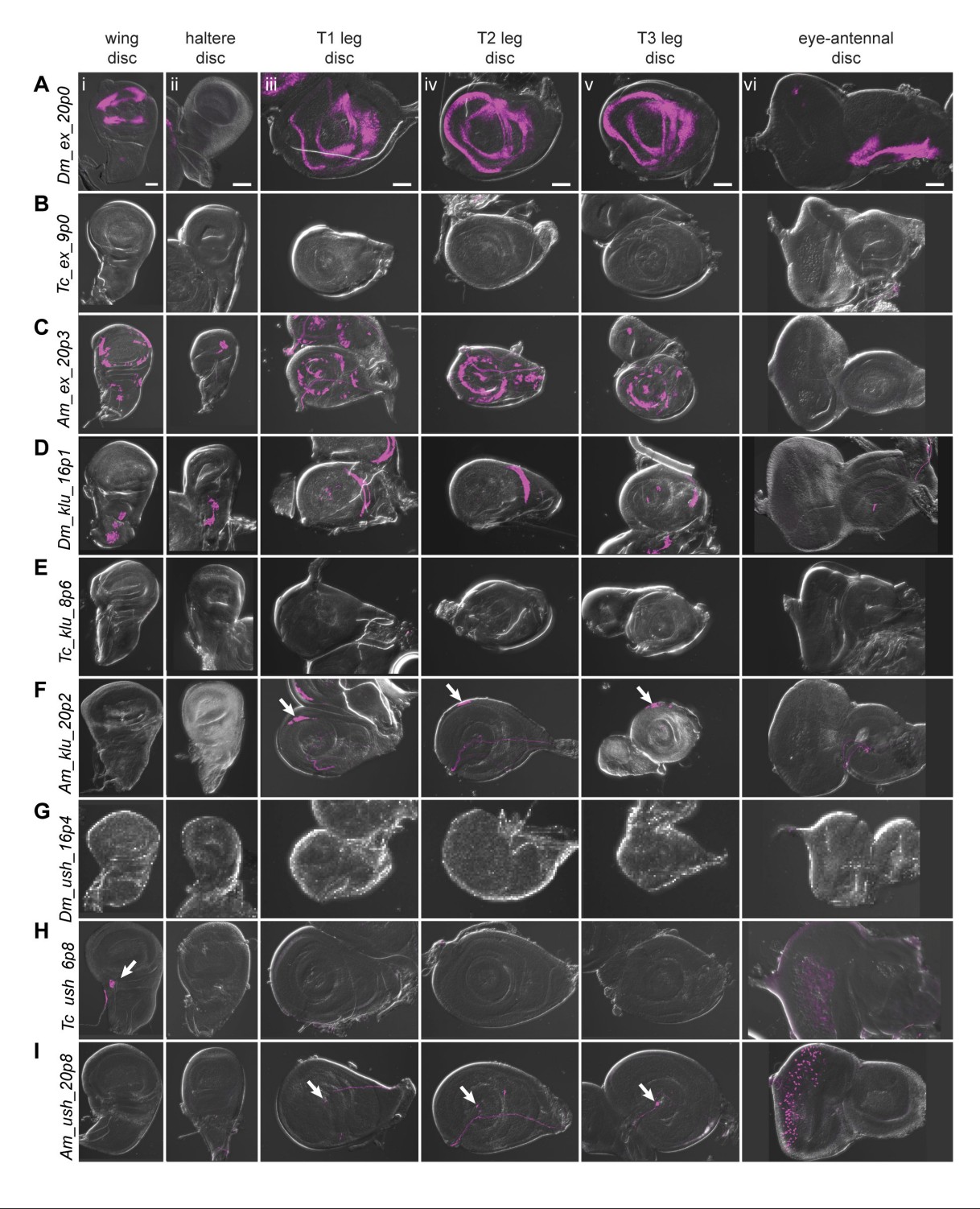

**Figure 6.** Reporter gene expression for tested *ex*, *klu*, and *ush* predicted enhancer sequences. Each row shows expression for the indicated construct in (i) wing discs, (ii) haltere discs, (iii) T1 (prothoracic) leg discs, (iv) T2 (mesothoracic) leg discs, (v) T3 (metathoracic) leg discs, and (vi) eye-antennal discs (with eye portion to the left). Positive results were obtained by the enhancers associated with the *ex* locus of *D. melanogaster* (wing, legs, antenna, A) and *A. mellifera* (wing, haltere, legs, C); the *klu* locus of *D. melanogaster* (wing, haltere, legs, D) and *A. mellifera* (legs, arrows in F); the *ush* locus of *T. castaneum* (wing, arrow, H (i); eye, H (vi)) and *A. mellifera* (legs, arrows I (iii, iv, v); eye, I (vi)). Enhancer activities were visualized by UAS-tdTomato that was included in the reporter construct. Scale bar is 50 µm for each column.

The online version of this article includes the following figure supplement(s) for figure 6:

*Figure 6 continued on next page*

**Figure supplement 2.** Additional expression observed in selected transgenic reporter lines.

orthologous *hth* locus, Aa_hth_35p9, located in the first intron (***Figure 7—figure supplement 1***). This sequence failed to display activity in imaginal discs (***Figure 7A***).

*T. castaneum* had 23 *hth*-locus predictions. We selected a high-scoring (albeit not the highest-scoring) sequence, Tc_hth_15p5, due to the similarity of its position to that of the *D. melanogaster* enhancer—in the fifth intron—and overlapping FAIRE peak (***Figure 7—figure supplement 1***; ***Lai et al., 2018***). The Tc_hth_15p5 reporter showed activity in the presumptive notum region of the wing disc and in the proximal region of the leg discs, resembling both the endogenous *D. melanogaster hth* expression pattern and that of the *GMR46D04* enhancer (***Figure 7B***, compare with ***Figure 5F***). No enhancer activity was observed in the haltere or eye-antennal discs (***Figure 7B***). In the pupal stage, Tc_hth_15p5 exhibited active expression along the thorax, corresponding to adult *hth* expression in *D. melanogaster* (***Figure 7—figure supplement 2***, arrows) (***Aldaz et al., 2005***). No *hth* predictions were obtained for *A. mellifera*.

## Ubx

The classic homeotic gene *Ultrabithorax* (*Ubx*) regulates tissue identity in the thoracic and abdominal segments (***Lewis, 1978***). Of six SCRMshaw predictions in the *D. melanogaster Ubx* locus, we noted that one, Dm_Ubx_36p1, overlaps a cluster of known enhancers in the third intron centered on *Ubx_GMR39A02* and *Ubx_abx6.8* (***Figure 7—figure supplement 1***). *Ubx_GMR39A02* mirrors the haltere activity of *Ubx*, but also displays ectopic activity in the pouch and notum regions of the wing disc (***Figure 5H***; ***Jory et al., 2012***). Similarly, *Ubx_abx6.8* also drives both native and ectopic expression in the imaginal discs (***Simon et al., 1990***).

There were eight predictions in the *Ubx* locus of the *A. aegypti* genome (***Figure 7—figure supplement 1***). We selected sequence Aa_Ubx_26p0, as it was both the highest scoring prediction and was in the third intron, similar to its putative *D. melanogaster* counterparts. Although direct reporter expression from Aa_Ubx_26p0 was too weak to observe directly, use of the lineage-tracing G-TRACE system confirmed widespread activity in all leg discs (***Figure 7C***).

We chose two *T. castaneum* sequences for testing, out of 10 predictions: Tc_Ubx_17p4, which overlapped well with an accessible chromatin region in the third thoracic epidermal tissue and the central nervous system (***Lai et al., 2018***), and Tc_Ubx_19p9, which had the highest local prediction score but only overlapped with a chromatin region accessible predominantly during early embryogenesis (***Figure 7—figure supplement 1***). Tc_Ubx_17p4 displayed activity in all three leg discs (***Figure 7D***). The expression in T2 and T3 leg discs corresponds to endogenous *Ubx* expression, whereas T1 leg disc expression appears to be ectopic. Tc_Ubx_19p9, predicted using the 'haltere' training set, did not drive clear expression in the haltere disc but did drive G-TRACE expression in the wing disc in the adepithelial adult myoblast cells, as well as direct reporter expression in the peripodial membrane of the eye disc. Very limited expression was also observed in the leg discs when using G-TRACE (***Figure 7E***, ***Figure 6—figure supplement 2J***).

The *A. mellifera* genome had 11 predictions at the *Ubx* locus (***Figure 7—figure supplement 1***). Am_Ubx_37p2 was chosen based on a high SCRMshaw score, although it falls within the fourth, rather than the third, intron. Am_Ubx_0p39, on the other hand, was in the corresponding third intron location. Am_Ubx_0p39 had activity in the pouch region of the wing and haltere discs, as well as in the proximal region of leg discs (***Figure 7F***). Am_Ubx_37p2 drove expression in specific portions of the wing and haltere discs (***Figure 7G***). Although *Ubx* is not expressed in the *D. melanogaster* wing disc (i.e. the forewing of the fly), wing activity has been observed previously with tested *Ubx* enhancer fragments (e.g. ***Jory et al., 2012***). Moreover, *Ubx* is expressed in both the forewing and hindwing discs in honeybees, making the reporter expression we observed driven by the two predicted *A. mellifera* enhancers consistent with their potential native activities (***Prasad et al., 2016***).

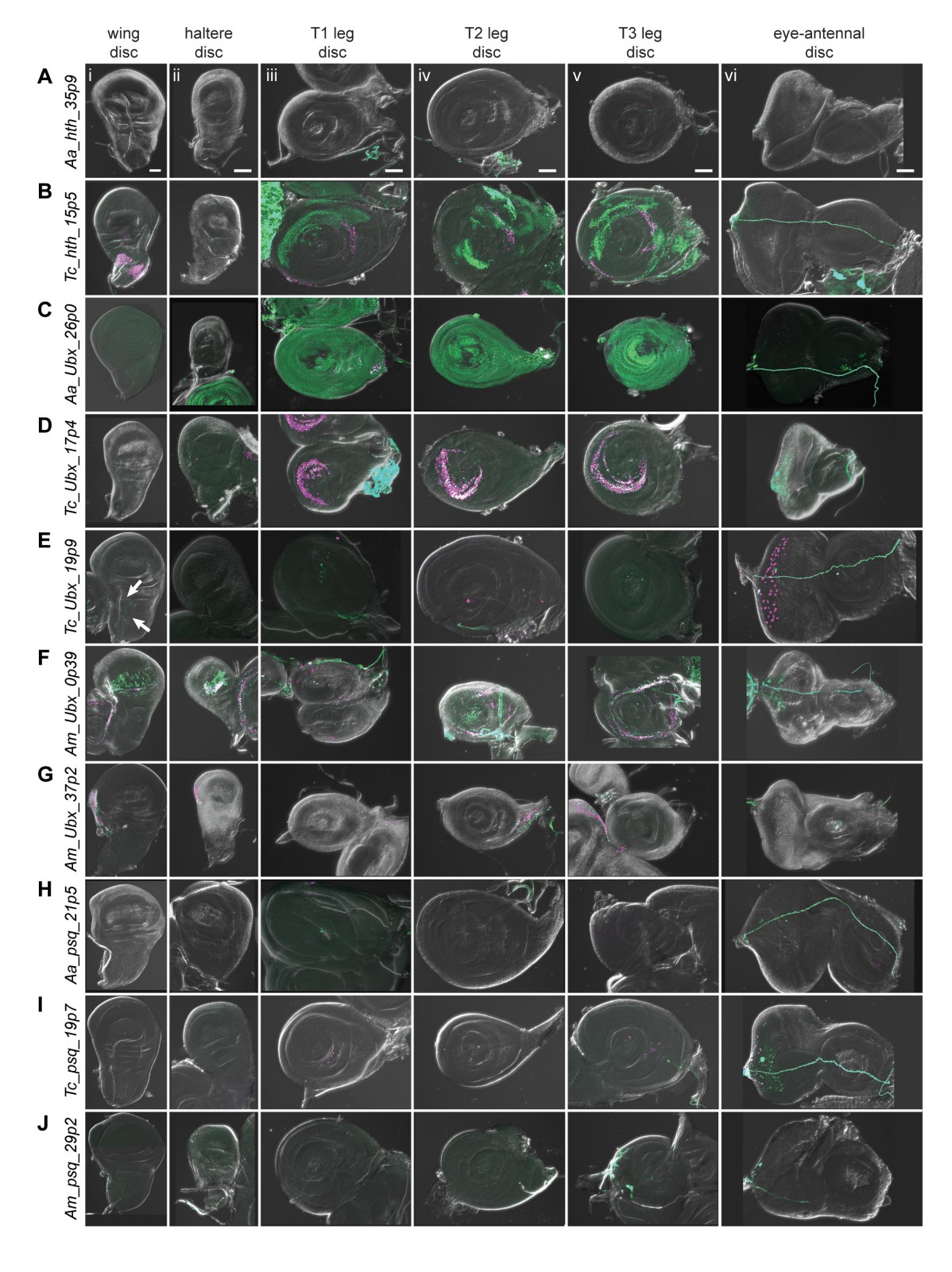

**Figure 7.** Reporter gene expression for tested *hth*, *Ubx*, and *psq* predicted enhancer sequences. Each row shows expression for the indicated construct in (i) wing discs, (ii) haltere discs, (iii) T1 (prothoracic) leg discs, (iv) T2 (mesothoracic) leg discs, (v) T3 (metathoracic) leg discs, and (vi) eye-antennal discs (with eye portion to the left). Positive results were obtained by the enhancers associated with the *hth* locus of *T. castaneum* (the notum portion of the wing disc, legs, B); the *Ubx* locus of *A. aegypti* (legs, C), *T. castaneum* (legs, D) (myoblast cells in the wing disc, arrows, E (i); legs, eye, E (iii–vi)), and *A.*

*Figure 7 continued on next page*

*Figure 7 continued*

*mellifera* (wing, haltere, legs, F) (wing, haltere, G); the *psq* locus of *T. castaneum* (legs, eye, I). Enhancer activities were detected by the G-TRACE system; magenta represents direct enhancer activity detected by dsRed expression, while green indicates lineage-based GFP expression. Scale bar is 50 µm for each column.

The online version of this article includes the following figure supplement(s) for figure 7:

**Figure supplement 1.** Open chromatin data for predictions in the *hth, Ubx,* and *psq* loci.

**Figure supplement 2.** Reporter gene expression observed in embryos.

## psq

We chose *pipsqueak* (*psq*), a transcription factor involved in Polycomb group gene silencing (**Huang et al., 2002**), and its orthologous loci as the final targets for enhancer validation. There were four predicted *psq* enhancers in *D. melanogaster*, one of which, *Dm_psq_25p6*, overlaps known enhancer *psq_GMR41E12* (**Figure 7—figure supplement 1**). This enhancer is located within the second *psq* intron and drives expression in all imaginal discs (**Figure 5J**; **Jory et al., 2012**). The *A. aegypti* genome had four predictions for the *psq* locus (**Figure 7—figure supplement 1**); we chose *Aa_psq_21p5*, located in the second intron and with the highest local SCRMshaw score, for validation (**Figure 7—figure supplement 1**). However, *Aa_psq_21p5* did not have observable imaginal disc expression (**Figure 7H**).

*T. castaneum* had only two predictions for the *psq* locus, both within the third intron (**Figure 7—figure supplement 1**). *Tc_psq_19p7*, which was chosen for validation due to its higher score, drove expression in the eye discs and in a very limited number of cells in the leg discs (**Figure 7I**).

From the *A. mellifera* genome, we selected the only prediction for the *psq* locus, *Am_psq_29p2* (**Figure 7—figure supplement 1**). *Am_psq_29p2* reporter activity was negative in all tissues assayed (**Figure 7J**).

## Embryonic activity

Although our SCRMshaw predictions were targeted toward imaginal disc activity, activity was also observed in embryos for many of the reporter lines (**Figure 7—figure supplement 2**). Analysis of this activity was complicated by the fact that our reporter lines, while not having any basal activity in imaginal discs (**Figure 6—figure supplement 2A–E**), displayed reporter gene expression in several tissues including hemocytes, caudal visceral mesoderm, and the proventriculus, even in the absence of a putative enhancer sequence (**Figure 7—figure supplement 2A–D**). Similar expression is seen in control embryos of the G-TRACE line alone, i.e., even in the absence of a Gal4 driver (data not shown), suggesting that the observed basal activity might be coming from the UAS construct. In those lines that had reporter gene expression in other tissues (**Figure 7—figure supplement 2I–S, X, Y**), most of that expression was not clearly associated with the expected endogenous expression of the predicted target gene, although the complex embryonic expression patterns of these genes make a definitive assessment difficult. Moreover, for most of the species, we do not currently know the expression patterns of either the gene in its native species (as opposed to the expression of its *Drosophila* ortholog) or the expression patterns of other nearby potential target genes. Further analysis, including additional control experiments, use of different reporter vectors, and assessment of gene expression patterns in each of the relevant species will be necessary before drawing final conclusions as to embryonic enhancer activity.

## An insect regulatory annotation resource

Taken together, the results from our simulations, our comparisons to open chromatin regions, and our in vivo validation experiments demonstrate that SCRMshaw is remarkably effective at predicting regulatory sequences across a wide range of insect species. To facilitate access to our SCRMshaw-based regulatory annotations, we created a database with the results from our predictions using all training sets and all completed species. This database, which is freely accessible as part of the REDfly insect regulatory annotation site (**Keränen et al., 2022**, http://redfly.ccr.buffalo.edu), contains processed final prediction data and can be searched and filtered by gene, *D. melanogaster* ortholog, training set, enhancer location, and various other criteria. We will continue to add to this database as additional species and training sets are run through our annotation pipeline.

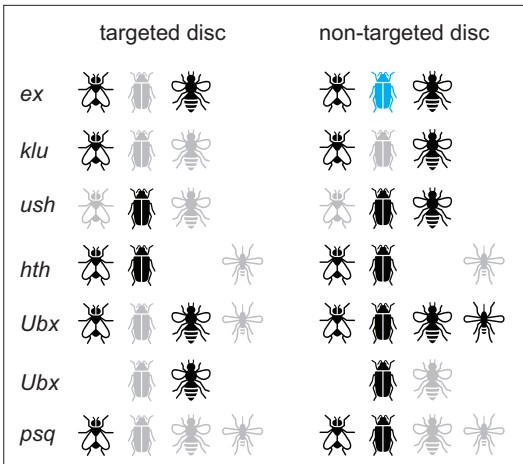

**Figure 8.** Summary of in vivo validation results. Results are shown for *D. melanogaster, T. castaneum, A. mellifera,* and *A. aegypti.* Black, positive for expression; gray, negative for expression; blue, expression observed in pupae but not in larvae.

Image credits: The insect silhouettes were obtained from The Noun Project (https://thenounproject.com/), artist Georgiana Ionescu, under a Creative Commons CC-BY 3.0 license.

## Discussion

The resource we introduce here is part of an ongoing effort to provide an initial regulatory annotation for all sequenced insects. These annotations are not complete, as we currently lack well-curated training sets for many tissues, including key embryonic tissues such as the central nervous system and many non-embryonic tissues. We aim to generate the necessary additional training sets over time, and add these new annotations to those presented here, along with comprehensive annotation for additional species. As a predictive pipeline, SCRMshaw is subject to the usual tradeoffs of sensitivity versus specificity in generating results. Sensitivity is difficult to assess, as there is no 'complete' known set of enhancers for any organism, and it is currently impossible to make an accurate estimate of what the yield should be for any of our training models. Similarly, without comprehensive in vivo testing using multiple conditions and methods, an accurate false-positive rate cannot be computed. However, we presented here several lines of evidence demonstrating that true positives significantly outweigh false positives: we non-randomly predict multiple enhancers for specific subsets of loci; we predict enhancers in orthologous loci at a rate significantly higher than random expectation; our predictions are highly enriched for regions of accessible chromatin; and we achieve a high rate of validation using in vivo reporter gene assays.

## Validation success rates

The results from the in vivo validation experiments are summarized in *Figure 8*. Overall, 17/22 (77%) of tested sequences revealed imaginal disc activity, consistent with our previous SCRMshaw success rates. 59% of these (10/17), or 45% of the total (10/22), had the correct target specificity of wing and/or haltere discs. This is again consistent with previous SCRMshaw experience, which shows that functional enhancers are predicted at a higher success rate than enhancers with specific targeted activity (*Kantorovitz et al., 2009*; *Kazemian et al., 2011*).

These numbers may in fact underestimate the success rate of our predictions, for several reasons. One, all of the testing was performed in transgenic *D. melanogaster*, despite the putative enhancers being from three additional species. Reduced efficiency of certain transcription factor or cofactor binding, reduced enhancer-promoter compatibility, or other species-specific differences may lead to elevated false-negative results. Furthermore, the unique imaginal disc mode of adult epithelial development in *D. melanogaster* (*Cohen, 1993*; *Svácha, 1992*) might have prevented some enhancers of other species from working properly in *D. melanogaster* imaginal discs, likely producing additional false-negative results. Evaluating enhancer activities in the native species will allow us to address the degree of false negatives produced by the cross-species setting. Two, predictions for *ex, klu*, and *ush* were made using a less-effective SCRMshaw post-processing algorithm and, for *A. mellifera*, a less-complete genome build. This may have led to suboptimal candidate enhancer selection. Three, it is possible that some of our predicted enhancers act as silencers rather than enhancers. Silencers, which attenuate rather than promote gene expression, are less well understood than enhancers, but in at least some instances have identical sequence characteristics (and in fact can act simultaneously as enhancers in some tissues and silencers in others) (*Halfon, 2020*; *Segert et al., 2021*). Our reporter gene assay was not designed to detect silencers, which would therefore appear as false-positive predictions. Indeed, an intriguing possibility is that some of our sequences that had reporter gene activity in non-targeted discs (e.g. leg discs) act as enhancers in those tissues but as silencers in the targeted wing discs. Alternatively, identified enhancers that drove expression in multiple imaginal

discs may simply represent pleiotropic enhancers. Enhancer pleiotropy is not uncommon (*Laiker and Frankel, 2022*; *Sabarís et al., 2019*), and moreover, there is overlap in the enhancer content of our training sets for the various discs. There are many gene expression and developmental similarities among the discs, and likely significant regulatory overlap; chromatin profiling experiments have found that wing, leg, and eye discs share the majority of their regions of accessible chromatin (*Lai et al., 2018*; *McKay and Lieb, 2013*). Additional experiments will be necessary to distinguish between pleiotropic enhancers, dual-function silencers/enhancers, and other possibilities.

## Redundant enhancers

Almost all of our training sets predicted multiple enhancers in certain loci at rates exceeding random expectation, consistent with the noted prevalence of redundant or 'shadow' enhancers in numerous plant and animal species (reviewed by *Kvon et al., 2021*). The few training sets that did not may reflect types of tissues or regulatory processes for which having shadow enhancers is unusual, or may be indicative of poorly performing training sets. Indeed, the sole training set that did not suggest the presence of shadow enhancers in either mosquito species, *adult_PNS*, also performed poorly by other metrics, such as failing to produce predictions in common over multiple species and scoring 'poor' using our *pCRM_eval* training set evaluation tool (*Asma and Halfon, 2019*).

Although the evolutionary origins of shadow enhancers are not well understood, several studies suggest that at least one important contribution of shadow enhancers is to provide phenotypic robustness during development, particularly during stress conditions (*Frankel et al., 2010*; *Perry et al., 2010*; *Perry et al., 2011*; *Osterwalder et al., 2018*; *Antosova et al., 2016*; *Sagai et al., 2017*). In some cases, a mechanistic basis for this robustness has been suggested by the finding that different groups of transcription factors act on individual members of a shadow enhancer set (*Waymack et al., 2020*; *Cannavò et al., 2016*). Further support for this idea can be found in the fact that shadow enhancers appear to have limited sequence similarity (*Waymack et al., 2020*; *Barth et al., 2020*). However, our successes with SCRMshaw have demonstrated that enhancers can have little overt sequence similarity but still rely on the same underlying subsequence (and presumably, transcription factor binding) model. As the potential shadow enhancers we identify are predicted using the same training set, they are likely to be functioning using similar, rather than independent, regulatory mechanisms. Thus, shadow enhancers appear to come in at least two flavors: those that use similar mechanisms, such as those identified here, and those that use different mechanisms, such as the *Krüppel* enhancers studied by *Waymack et al., 2020*.

## Enhancer evolution at large divergence distances

SCRMshaw's ability to find putatively 'orthologous' enhancers—i.e., enhancers in orthologous loci predicted using the same regulatory model—not only substantiates the non-random nature of our prediction approach, but also opens the door to exciting studies of enhancer evolution over large divergence ranges. We previously illustrated the power of such an approach with an analysis of a small number of enhancers in just two to three highly diverged species (*Kazemian et al., 2014*). However, the greatly increased number of species for which we now have regulatory predictions should allow us to follow the evolution of specific enhancers over their entire divergence range. Although the number of common loci tails off rapidly as the number of species considered increases, we suspect that this small (albeit statistically significant) number understates the true results. For one, only a limited number of species have been evaluated for common predictions so far, and these are not evenly distributed along the phylogenetic spectrum of sequenced species. Also, our analysis depends wholly on the presence of recognized *D. melanogaster* orthologous genes to define the common loci. This in turn is dependent on several factors, including the sensitivity and accuracy of our ortholog-calling pipeline, the reliability of the protein annotations we use in that pipeline, and on accurate target gene assignments. Each of these has known sources of error. For example, our ortholog-calling pipeline currently does not disambiguate multiple paralogs (see Methods), and ortholog detection has been shown to be sensitive to the method used for generating protein annotation, in particular when different annotation methods have been applied to different species in the comparison (*Weisman et al., 2022*). Moreover, our target gene assignments are presently based solely on closest-gene relationships, a method which is known to mis-assign a fair number of enhancer-gene target pairings (e.g. *Sanyal et al., 2012*; *Hafez et al., 2017*; *Chua et al., 2022*; *Qin et al., 2022*) and which does

not take into account complexities in genome architecture such as nested genes, long non-coding RNAs, promoter competition, and the like (which are common in *D. melanogaster*, e.g. *Crosby et al., 2015*; *Matthews et al., 2015*). Developing effective and scalable solutions to these issues will be an important goal for future work.

As the number of species in our prediction database with fully mapped orthologs grows, it will become possible to ask increasingly sophisticated questions about the nature of enhancer evolution. For instance, we will be able to determine whether the numbers of putatively orthologous enhancer predictions follow phylogenetic relationships and degree of sequence divergence, and whether certain loci are only found in common for specific groups of species. Also in need of further investigation will be to determine whether sets of common enhancers are restricted to certain functional Gene Ontology categories, and how this might vary with phylogenetic grouping.

### Ongoing regulatory annotation

While effective, SCRMshaw still has various limitations and aspects in need of improvement. Errors in genome assembly and insufficient repeat masking both appear to contribute to overly long predictions (multiple kilobases) that are unlikely to represent individual single enhancers (HA and MSH, unpublished observations). Poor assembly, while not having major effects on SCRMshaw overall, can significantly affect results at specific loci, as can inaccurate gene annotation (*Asma and Halfon, 2019*). Also requiring further investigation is how to best combine and weight the scores from the three individual SCRMshaw scoring methods of IMM, hexMCD, and PAC-rc. Individual SCRMshaw predictions should therefore be treated as just that—predictions—and appropriate validation experiments are recommended for any sequences of interest. The regulatory annotations presented here represent initial '1.0' versions of the regulatory genome. Like all genome annotations, these will continue to be revised as updated models become available, including addition of new training sets, confidence scores based on validation experiments, and improved genome builds. Nevertheless, the enrichment scores from our in silico experiments and the true-positive rates from our in vivo reporter gene experiments suggest that overall, true-positive rates for most training sets are between 50% and 85%, with most likely having success rates exceeding 70%. Note that even this lower bound of a 50% true-positive rate means that one out of every two predictions is correct—a more than acceptable rate to encourage follow-up experiments for predicted enhancers in organisms of interest. The catalog of predicted enhancers we introduce here, spanning 33 insect species and growing, should thus be a useful resource for both large-scale and small-scale studies of the insect regulatory genome.

## Methods

### Datasets

Sequence and annotation for *D. melanogaster* were obtained from FlyBase (*Gramates et al., 2022*). For other species, wherever possible, genome sequence (FASTA) and annotation (GFF) files were downloaded from NCBI at https://www.ncbi.nlm.nih.gov/datasets. Otherwise, the genome sequences and annotations were obtained from the primary literature or directly from the data generators (see *Table 1* for references). Version numbers for genomes and annotations are provided in *Table 1*.

### Scripts

A permanent archive of this code can be found at Zenodo: https://doi.org/10.5281/zenodo.13821366. All scripts used for the analyses described in this paper can be found in the GitHub repository Asma_etal_2024_eLife at https://github.com/HalfonLab/Asma_etal_2024_eLife, (copy archived at *Asma and Halfon, 2024*).

### SCRMshaw

A detailed SCRMshaw protocol can be found at *Asma et al., 2024*.

Genome files were masked using Tandem Repeat Finder (*Benson, 1999*) using parameters 2 7 7 80 10 50 500 -m -h. Genome and annotation files were then assessed using our *preflight* script. Any sequence scaffolds not containing annotated genes were removed before passing the genome sequence to the main SCRMshaw program. SCRMshaw was run using the 'HD' version, which scans the genome with a 500 bp window sliding in 10 bp increments, as described in *Asma and Halfon,*

*2019*. The `--thiwt` parameter was set to 5000, and `--lb` to [0,10,20,30,…,240] for each of 25 instances, respectively. All other parameters were kept at default values.

SCRMshaw makes use of three underlying statistical models, described briefly in *Kazemian et al., 2014*, and in detail in *Kantorovitz et al., 2009*; *Kazemian et al., 2011*. *HexMCD* trains a fifth-order Markov chain on all 6-mers and their one-away mismatches in the training and background sequence sets, respectively, while *IMM* trains an interpolated Markov model that combines Markov chains of all orders from 0 to 5. For both of these methods, the score for a sequence is the log-likelihood ratio of the model trained on the training sequences divided by the model trained on the background sequences. *PAC-rc* quantifies the overrepresentation of 6-mers in the training versus background sets, assuming a Poisson distribution of word counts. Our previous work has shown that each method is effective, and each tends to have sensitivity for a different group of enhancers. Roughly three-quarters of all predicted enhancers were identified by just a single method (see *Figure 2—source data 2*), although there is a strong correlation between high-ranking predictions and prediction by more than one method. The best results are obtained from combining the output of all three methods.

SCRMshaw can be downloaded at https://github.com/HalfonLab/SCRMshaw_HD, (copy archived at *Asma and Halfon, 2023*).

## Post-processing

The top 5000 hits were extracted using scripts *Generate_top_N_SCRM-hits.pl* and *concatenatenatin gOffsetResults.sh* and passed to *postProcessingScrmshawPipeline.py* with *-num* = '5000' and *-topN* = 'Median'. Post-processing was performed essentially as previously described (*Kazemian and Halfon, 2019*; *Asma and Halfon, 2019*), with the following modifications (*Figure 1—figure supplement 1*): before evaluating each 10 bp region, scores from each of the 25 individual SCRMshaw instances were assessed and any 500 bp window whose score was below the value of the 5000th ranked score was eliminated by having its score reset to zero (*Figure 1—figure supplement 1B*). The 'elbow' point of the SCRMshaw score curve of the 5000 top scores from each instance was then determined (*Figure 1—figure supplement 1C*), and any scores below the elbow point were reset to zero (*Figure 1—figure supplement 1D*; gray boxes). This is a key modification to our previous SCRMshawHD protocol (*Asma and Halfon, 2019*) and reduces the median prediction size by preventing concatenation of multiple adjacent low-scoring windows. Only after these two rounds of score evaluation were all windows grouped together (*Figure 1—figure supplement 1E, F*) and subjected to peak calling on 10 bp intervals (*Figure 1—figure supplement 1G*). Final 'top predictions' were then any peaks with an amplitude above the selected amplitude threshold (elbow point of amplitude curve, represented by a red dot in *Figure 1—figure supplement 1H*), following the peak-calling step.

All scripts are available at https://github.com/HalfonLab/.

## Orthology mapping

The final SCRMshaw output from the above steps was used as input to our orthology mapping pipeline. For each species, a FASTA-formatted file of all annotated proteins was downloaded from NCBI (https://www.ncbi.nlm.nih.gov/datasets). The annotated proteins from *D. melanogaster* plus each individual other species were used as input to *Orthologer* (*Kuznetsov et al., 2023*) to obtain the *Drosophila* ortholog for each protein (when existing). Our approach was designed to be minimally restrictive in that we did not enforce a one-to-one ortholog mapping; in cases of likely paralogs, we considered all of the paralogs as a potential result. Details on the orthology mapping protocol can be found in *Asma et al., 2024*.

## Evaluating the number of predictions per locus

For each training set, BEDTools 'merge' was used to remove any overlapping predictions (*Quinlan and Hall, 2010*). The results were then permuted 1000 times using BEDTools 'shuffle', with coding regions excluded. BEDTools 'closest' was used to assign upstream and downstream flanking genes to each of the permuted predictions, with the following parameters: the '-io' flag was enabled to ignore any overlaps between predictions and genes, and '-D ref -id' and '-D ref -iu' were used to obtain the closest 5' and 3' genes (with respect to chromosome coordinates), respectively. A custom Python script, *checkSameLocus_ForSimulations.py* (see https://github.com/HalfonLab/Asma_etal_2024_eLife), was used to calculate the number of predictions per locus for both the real and permuted

results. For this purpose, 'locus' was defined as the entire region between the left and right flanking genes for each SCRMshaw prediction; e.g., for a prediction between two genes in a head-to-tail orientation we take the region between the 3' end of the upstream gene and the transcription start site of the downstream gene. If a prediction is within an intron, we define the locus as the entire span of the enclosing gene. If a prediction overlaps two genes, the locus is considered to be the entire span of the two genes. Average intergenic region sizes for the genomes were estimated using the distances between neighboring genes, discarding any nested or overlapping gene pairs, as provided by the SCRMshaw *preflight* script.

Significance was assessed by calculating the empirical p-value, defined as the number of real loci with a number of predictions greater than the maximum number obtained from the 1000 simulations, for each locus containing at least one SCRMshaw prediction.

## Evaluating the number of predictions in common across species

To determine the expected number of common predictions—i.e., predictions in orthologous loci—across species, we utilized SCRMshaw predictions for 16 species. Each set of predictions was sorted, merged, permuted, and mapped to new loci as described above for 'Evaluating the number of predictions per locus'. The permuted predictions were used as input for the script '*checkSameLocus_ForSimulations_crossSpecies.py*' (see https://github.com/HalfonLab/Asma_etal_2024_eLife). This script identifies the *Drosophila* orthologs of the nearest flanking genes for each species and calculates the number of common loci flanking the simulated SCRMshaw predictions for 5, 10, 11, 12, 13, 14, 15, and 16 species. For each species, the process was repeated for a total of 360 permutations. The mean and standard deviation of the permuted results were then used to calculate a z-score for each training set. We considered training sets with z-score ≥1.645 to be significant (p<0.05, not corrected for multiple testing).

## Overlap between SCRMshaw predictions and open chromatin regions

Open chromatin data were obtained from the following sources:

*D. melanogaster*: Data for *D. melanogaster* were downloaded from GEO and consisted of ATAC-seq and FAIRE-seq data from accessions GSE101827, GSE38727, and GSE118240. These assays were performed using blastoderm embryos, eye-antennal discs, wing discs, haltere discs, leg discs, third instar central nervous system, and wing, leg, and haltere pharate appendages (*Bozek et al., 2019*; *McKay and Lieb, 2013*; *Jacobs et al., 2018*).

*T. castaneum*: FAIRE-seq data for three stages of embryogenesis, larval central nervous system, and larval second and third thoracic epidermal tissues were downloaded from GEO (GSE104495) (*Lai et al., 2018*). The FAIRE profiles were remapped to the version 5.2 of the *T. castaneum* genome (Tcas5.2) for this study. The remapped FAIRE profiles are available on iBeetle-Base (https://ibeetle-base.uni-goettingen.de/, *Dönitz et al., 2018*).

*A. gambiae*: ATAC-seq data for adult midgut and salivary gland were downloaded from GEO (GSE152924) (*Ruiz et al., 2021*).

*D. plexippus, J. coenia, H. himera,* and *V. cardui*: ATAC-seq data for larval forewing and hindwing tissues at stage M5 were provided by Anyi Mazo-Vargas and Robert Reed (*Mazo-Vargas et al., 2022*).

The overlap between SCRMshaw predictions and open chromatin regions was determined using BEDTools 'intersect' with parameters -wa -u -f 0.1 such that sequences needed to overlap at least 10% of their length and were considered a single overlap in the event that more than one open chromatin peak overlapped a prediction.

To assess significance, the SCRMshaw predictions were permuted 500 times using BEDTools 'shuffle' and the overlaps with open chromatin regions assessed as above. The mean and standard deviation of the permuted results were then used to calculate a z-score for each training set. We also calculated a 'fold enrichment' score by dividing the observed number of overlapping regions by the expected number, to provide a sense of effect size in addition to statistical significance.

## Reporter constructs and transgenic *Drosophila*

Sequences for reporter gene analysis, including attL1 and attL2 sites, were synthesized de novo and cloned into pUC57 Kan-r (GenScript, Piscataway, NJ, USA) as entry vectors suitable for Gateway cloning (*Katzen, 2007*). Gateway LR recombination was then used to move the sequences into

piggyPhiGUGd (*hth, Ubx,* and *psq* lines) and piggyPhiGUGd-TomatoI (*ex, klu,* and *ush* lines) (*Deem et al., 2024*) (these reporter vectors are available from the Drosophila Genomics Resource Center, https://dgrc.bio.indiana.edu). Transgenic flies were generated by BestGene (Chino Hills, CA, USA) using PhiC31 recombination and the attP2 third chromosome insertion site, and are available on request. piggyPhiGUGd lines were subsequently crossed to G-TRACE for visualizing enhancer activities (*Evans et al., 2009*).

For each construct, imaginal discs from at least six larvae were dissected and mounted for direct fluorescence visualization using a Zeiss Axio Imager M2 microscope with ApoTome 2. Embryos were fixed and stained using standard *Drosophila* methods using anti-dsRed (Clontech) and visualized using the ABC-HRP kit (VectorLabs, Newark, CA, USA).

## Database implementation

The SCRMshaw results database is implemented as part of REDfly (RRID:SCR_006790; *Keränen et al., 2022*), a MariaDB-based database hosted on a private OpenStack cloud infrastructure maintained by the University at Buffalo Center for Computational Research. As part of an ongoing transition of REDfly to a more modern software architecture, backend functions are implemented in Node.JS and Python, while frontend components utilize React.JS and Next.js. GraphQL is used as the query language. REDfly is licensed under a Creative Commons Attribution-NonCommercial-NoDerivatives License v4 International (CC BY-NC-ND 4.0) and its underlying source code under a GNU General Public License v3 (GNU GPL 3.0).

## Acknowledgements

We thank Bob Reed, Anyi Mazo-Vargas, and Tom Williams for sharing data and results, Jack Leatherbarrow for technical support, members of the Halfon and Tomoyasu labs for helpful discussion and advice, and Tom Williams for comments on the manuscript. SCRMshaw analyses were run using the resources of the University at Buffalo Center for Computational Research. The Center for Bioinformatics and Functional Genomics at Miami University provided instrumentation and technical support.

## Additional information

### Funding

| Funder | Grant reference number | Author |
| --- | --- | --- |
| National Science Foundation | IOS-1557936 | Yoshinori Tomoyasu |
| National Institutes of Health | U24 GM142435 | Marc S Halfon |
| National Institute of Food and Agriculture | 2019-67013-29354 | Yoshinori Tomoyasu Marc S Halfon |
| Miami University | Faculty Research Grants Program | Yoshinori Tomoyasu |

The funders had no role in study design, data collection and interpretation, or the decision to submit the work for publication.

### Author contributions

Hasiba Asma, Software, Investigation, Methodology, Writing – original draft; Ellen Tieke, Investigation, Writing – original draft; Kevin D Deem, Jabale Rahmat, Tiffany Dong, Investigation; Xinbo Huang, Software; Yoshinori Tomoyasu, Conceptualization, Supervision, Funding acquisition, Writing – review and editing; Marc S Halfon, Conceptualization, Software, Supervision, Funding acquisition, Methodology, Writing – original draft, Writing – review and editing

### Author ORCIDs

Hasiba Asma (ID) https://orcid.org/0000-0002-9304-2685
Yoshinori Tomoyasu (ID) https://orcid.org/0000-0001-9824-3454

Marc S Halfon ⓘ https://orcid.org/0000-0002-4149-2705

Reviewer #1 (Public review): https://doi.org/10.7554/eLife.96738.3.sa1
Reviewer #3 (Public review): https://doi.org/10.7554/eLife.96738.3.sa2
Author response https://doi.org/10.7554/eLife.96738.3.sa3

## Additional files

### Supplementary files
• MDAR checklist

### Data availability

Data generated in this study has been deposited at Dryad at https://doi.org/10.5061/dryad.3j9kd51t0 in both the standard SCRMshaw output format (modified BED) and in GFFv3 format. Data can also be obtained via the REDfly database at http://redfly.ccr.buffalo.edu. Software is available at https://github.com/HalfonLab/Asma_etal_2024_eLife, copy archived at *Asma and Halfon, 2024*.

The following dataset was generated:

| Author(s) | Year | Dataset title | Dataset URL | Database and Identifier |
|---|---|---|---|---|
| Asma H, Tieke E, Deem KD, Rahmat J, Dong T, Huang X, Tomoyasu Y, Halfon MS | 2024 | Data from: Regulatory genome annotation for 33 insect species | https://doi.org/10.5061/dryad.3j9kd51t0 | Dryad Digital Repository, 10.5061/dryad.3j9kd51t0 |

The following previously published datasets were used:

| Author(s) | Year | Dataset title | Dataset URL | Database and Identifier |
|---|---|---|---|---|
| Jacobs J, Aerts S | 2018 | The transcription factor Grainyhead primes epithelial enhancers for spatiotemporal activation by displacing nucleosomes [ATAC-seq] | https://www.ncbi.nlm.nih.gov/geo/query/acc.cgi?acc=GSE101827 | NCBI Gene Expression Omnibus, GSE101827 |
| McKay D, Lieb JD | 2013 | A common set of DNA regulatory elements shapes *Drosophila* appendages | https://www.ncbi.nlm.nih.gov/geo/query/acc.cgi?acc=GSE38727 | NCBI Gene Expression Omnibus, GSE38727 |
| Bozek M, Cortini R, Storti AE, Unnerstall U, Gaul U, Gompel N | 2019 | Genome-wide profiles of chromatin accessibility in spatially-restricted domains along the antero-posterior axis of *Drosophila* blastoderm | https://www.ncbi.nlm.nih.gov/geo/query/acc.cgi?acc=GSE118240 | NCBI Gene Expression Omnibus, GSE118240 |
| Lai Y, Deem KD, Borras-Castells F, Sambrani N, Rudolf H, Suryamohan K, El-Sherif E, Halfon MS, McKay DJ, Tomoyasu Y | 2017 | Enhancer identification and activity evaluation in the red flour beetle, Tribolium castaneum | https://www.ncbi.nlm.nih.gov/geo/query/acc.cgi?acc=GSE104495 | NCBI Gene Expression Omnibus, GSE104495 |
| Ruiz JL, Ranford-Cartwright LC, Gómez-Díaz E | 2020 | The regulatory genome of the malaria vector Anopheles gambiae: integrating chromatin accessibility and gene expression | https://www.ncbi.nlm.nih.gov/geo/query/acc.cgi?acc=GSE152924 | NCBI Gene Expression Omnibus, GSE152924 |

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

# Appendix 1

## Appendix 1—key resources table

| Reagent type (species) or resource | Designation | Source or reference | Identifiers | Additional information |
|---|---|---|---|---|
| Gene (*Drosophila melanogaster*) | expanded (ex) | FlyBase | FBgn0004583 | |
| Gene (*D. melanogaster*) | u-shaped (ush) | FlyBase | FBgn0003963 | |
| Gene (*D. melanogaster*) | klumpfuss (klu) | FlyBase | FBgn0013469 | |
| Gene (*D. melanogaster*) | homothorax (hth) | FlyBase | FBgn0001235 | |
| Gene (*D. melanogaster*) | pipsqueak (psq) | FlyBase | FBgn0263102 | |
| Gene (*D. melanogaster*) | Ultrabithorax (Ubx) | FlyBase | FBgn0003944 | |
| Gene (*Tribolium castaneum*) | gene-LOC657053 | iBeetleBase | TC012545 | |
| Gene (*T. castaneum*) | gene-LOC659918 | iBeetleBase | TC013689 | |
| Gene (*T. castaneum*) | gene-LOC103312803 | iBeetleBase | TC002783 | |
| Gene (*T. castaneum*) | gene-Hth | iBeetleBase | TC008629 | |
| Gene (*T. castaneum*) | gene-LOC660343 | iBeetleBase | TC003349 | |
| Gene (*T. castaneum*) | gene-Ubx | iBeetleBase | TC000903 | |
| Gene (*Apis mellifera*) | gene-LOC551519 | RefSeq | | |
| Gene (*A. mellifera*) | gene-LOC100577801 | RefSeq | | |
| Gene (*A. mellifera*) | gene-LOC100577692 | RefSeq | | |
| Gene (*A. mellifera*) | gene-LOC552079 | RefSeq | | |
| Gene (*A. mellifera*) | gene-psq | RefSeq | | |
| Gene (*A. mellifera*) | gene-ubx | RefSeq | | |
| Gene (*Aedes aegypti*) | AAEL001437 | VectorBase | | |
| Gene (*A. aegypti*) | AAEL020615 | VectorBase | | |
| Gene (*A. aegypti*) | AAEL013544 | VectorBase | | |
| Gene (*A. aegypti*) | AAEL011643 | VectorBase | | |
| Gene (*A. aegypti*) | AAEL021255 | VectorBase | | |
| Gene (*A. aegypti*) | AAEL014032 | VectorBase | | |
| Antibody | anti-dsRed (Rabbit polyclonal) | Clontech | Cat#632496 | (1:500) |
| Recombinant DNA reagent | piggyPhiGUGd (plasmid) | *Deem et al., 2024*, PMID:38698030 | | |
| Recombinant DNA reagent | piggyPhiGUGd-TomatoI (plasmid) | *Deem et al., 2024*, PMID:38698030 | | |

*Appendix 1 Continued on next page*

*Appendix 1 Continued*

| Reagent type (species) or resource | Designation | Source or reference | Identifiers | Additional information |
|---|---|---|---|---|
| Sequence-based reagent | Dm_ex_20p0 | This paper | | TTCCCAGAACAAACTTGTGGGGGGTGATTAGGTTTGGCAACAAAATAT TTTGCTAGTATTCCCTAATCATTTTTTTGAGTGAACCAAACTCGAAGAG CTCTACTCCCCTGGCCATCCACTTGTTGCCACTTCCATTCCAGCTTTG CGTCGACGACGTCGTCATTGATAGGCACTTATTCGGCCGCTGATGATT ATTATGATATTGTAGCTGCTGCTGCTGTGTTGTGGATTCGATGCTGAGG TGCCTCTATTCCATGGCCTCCTTCAACCTGCCTGCCTGCTTTTTTCATA ATTATTATTTTTCATCTTGCTGCTCTTCATTTTGTATGCAGGAATTCCAAT TTTTCGTTCGATGAAGTGTGTGTTGATTTCGCTGTTGTTTTTTCTCTGC CTTCTCGAGCACCGCCGACATGCCCTTGGGCCCTTCTGCTTGGCTCG GGTCGGAGCTATGTAGCGCGGTCCGGTACCGGTCTCGTCTTCGAGCA TCAGGCAATGGGCCTCTGACAACCTGACGTGTCGTCATCATCATCGTC TTCATCTGCTGGAGTCTCTGACTCTTGTTGATGTCAATGGGTTGCTTGT TTATTGCCTGACAAACTGACAGAAGTCTGGTCGGGGTCTCGATCCGATT TGAGCCCGATTTGGGACGCAAGAGGAGCGCTCCCTCTTGCATAGCCGA AAGTTCATTTAAAATTTTGAT |
| Sequence-based reagent | Dm_klu_16p1 | This paper | | GACCAGGCTGTTGCAGTTTCGTGTTGAAACCAGTTTGAATATATTTATTTT TATTTCCTGCGTCCCCTTCCCAATTTCTGTGGCCCTTTTAGGCGCCTCAG TTAGTCGGCAACGATAAGGCGGCAATGGTTTAATTTAGCTGCACCAGCGG CAGCAGCAGATGACGACAGGATCGTTGGGCCGGTCTACGTGCAACAGAA GTTGCTGCGCCGGCAGAAGCAACGGCAGCGGCAGCAACAGCAACAGCA AAACAAATGTGTCTGTATATCGCAGCTAAATTGACTTTGATCACGCGATCC CGAATCCCCCCCCCCCATTTGGTCCGAGTTATATGGCCGATTCCAGGTTGC AGGCTCCAGGTTCTCGGGCGCGGCCTTTTGTGGCACAAACGGAAGTATG CTAAGCAACTTGTTGCTGCCGCAAAGGCAAAGCAGCAAAAGCAGCTGAA GGTGTATATTGCAAAAATAATTACATTTGATTGTAAAAGGCCAGCGTCTCTA GGCTGGGGACTCGAGATCGGACCTCGGCCTGCCAGAGAAAAATGTGCAA CATGATTGCAGTTTACAGCCCCAGCAGCAGCAGCAGCAGCTAAAGCAGCA ACAACAACAGCAGCAGCAGCAGCAAAACCAACAGATAAAATGCATTTACA ATTGAATTTGAA |
| Sequence-based reagent | Dm_ush_16p4 | This paper | | GAATGTTGCTGCGGTGGCATGTTGTTGCTCGTCGAAGTTCAGCCGATGTT GCTGCTGCTGCTGCAGCTGCTGTTGCTGCCATTCCCACTATCAACCGATG GTAATCGAAGGAAGCGACATTTATGCAAATGCCAGGTTGTTTAATAAACGC AAATTATGAGCCCGGCAGCAACATGTTGCAGCAACAGTCGATGGCAGATT AGCGACATTCATACTTGCACTTGGGTCAATTTAAATTTGTGCAACAGTGGC AGCACGGCACGGCAGCAACTCCTTGCCGCAGCAGCAGCCGCTCCAGCA GCACATGAGATTGTGGGAGCAACAGGCAGCATTATTGTGTTCGGCCAAGA TCGCAATTGATCAGTGTGTTGGTGCTGGTGTTGGTGTTGCAGTTGCAGTT GCAGTTGCAGTTGCAGTCGCTGTTGCTGTTGCTGTTGCTACCCGACGACA ACAGTTGCTGCTGTGCTGGTGCTAGTGCTTGTGCTTGTGCTTGTGCTTGT GTTGCTGCTGATCAAGCGATCAAGCACCGCAGCCAAAAACAATCGGCGCT GAGCGTGCTCACACGAAATTTTCAAGTACTGCGACAATTTCCATGCCCCCA GCCGCTGCCTTGTTATCAGCGCGCCATGCAACAGCAACAGCGACTGCAGC ACAGGCAACAGCAGCAACACATCTCAACAGTTGCATCAATTGCTCAACATT GAACTCTGGGCATGGGCCCAGACGATCACCCCTCCTGGGGACCCCCTTC GGTCGCCCCTGCCCCAGTCCCTTGTATCATTTGCACGTTTTTTAATTAAGA CATCAAAA |
| Sequence-based reagent | Tc_ex_9p0 | This paper | | ATAGTTCTAAAGTTCTAAACTATTTGCAAATGTAAACAACGACCGACATTTT CAACATGTTCGGGTGTACGTCGCTTTGAATATGGAAAATGTGATTTTGTAG AGAAATTTGGTGGCGTAGCCGTTGCCAGCTCCAGTTTCTTAAGGCAGGC ATCTGGTAGCGCGCATCACAGCAGGCCGGGCCAGGTCAATAAAAAAATCG AGTCAGCCGTGGGCTAAAAAACGCGACTAAAAAAACAATGCAGAGCCGC GGTTAAAGACAGGTAGCCGAGCTAAGCGGTAGGGGAGAGGGGGATCAG GATCATTTGTATACTCGGGGAATGTGCCCGACCCGGTACACTCGATGCCA AAACGAACACGCCGACTTATACAGATGCCTATCTGACAATACGGACACTT TTAAAAAACACTTTTTCGGTTTTTAAAAGTTAAAGCCAACAAGCGCTGTG TTTTACACAATTCTTCCAATTTCCATTCCCAAGTTGCAAAAGTGAAACGT CCCAAAATATTTTGTCGCGCAAATCAAAAGTATATTTTATTCATGAGGAG CTCGTGTAATTTTTATGTAAATTTTAATTATTGATAACAAGGGACCATTGT TTGACGACACTTTCTTCGGCATGCGGAGCTCCTTGTTTTGT |
| Sequence-based reagent | Tc_klu_8p6 | This paper | | TTATGAGTTTGGTTATCGTCGGAATCGGGCATTCTTCCTTTGTATCCGAT TTTTAGGTAACAAGCTAGAAATTCCAGAGCTACACCTACGATCCATCAAG TCGGAGCCGTTCTAATTGGCCGCTCCTATCTATCGTCTGAAGGAGGCAG CAAGCAGCACACGAACACCTGGCTGCCAATGCACCTGATGCGGTCGTC CGTCGCTGCTATCAACTCACAATGTTTACTTGTGCTTACGCCAAAATTATG ACGAATATTAATGCGGCCTCGTACGCAGGCAGGCATGCGGCGTAATTACT ACCGGAGGGACCTTATCTCGAATTATTATGCAGCAAGCAGCAGTGAAAAA TAGCGCAACGCCTGCTGCACTCATCCAGATCTGACAAAGATGAAGACGT CGCTGACATCTTTATGATTGTGCTTTTATTGCACCTTTTCGCGGAATTCCG ACCATTCGAAGCGACCTTTGCGCTACGGAGAAGAGGAAATTTACACCGG GAGTTGACTTATGATGGGAGAACCACTCTCAACGAACGCAACTACTTTCC AGGAATATGAGAAGAGTGCTTAACTGACAAGTCCAAATTCGAACTTGAGGTTA |

*Appendix 1 Continued on next page*

*Appendix 1 Continued*

| Reagent type (species) or resource | Designation | Source or reference | Identifiers | Additional information |
|---|---|---|---|---|
| Sequence-based reagent | Tc_ush_6p8 | This paper | | TAAATAATCCAGACATGCATTGCATGTAAGTATCAGAGATACACGGTAAGAG TGGAGCTTTTCGAGAATCCGGAAACCGATCAGATAAGTTCTGAAAATGACT CGTCCACGAATAGATTTAGGATCGGAGCTGTTTTCCATTCGCCGAGATAAG TCGATAAGTTTCAATAAGTCCGAGTTCTGGCAACAGCCAGCACGGTACGG GCTGCCGCCGTCTTGGTTTCCAGTTTTCTCCAATGTCGTGGTATTAATCAG GGCGTTATCTCTAGCACATAAACACACGTATGTGTATGTGGGGCGATGTCG GTGGCGATACCGTTCCATGTGGGGGTGTAGCTGTTGGGGGTATACGGGC CTGTTCGCCGTCCGATAGCGCGAAAGATACGACCTGGAAGTAGGAAACG AGACAGCGAGATAAGAAAGTAATATGGCGGCTGCTGCAAAGAGATAACGA CTGATACGCGCCTGCACCTTTCCCGACCTGCAACTCTACGTGCCCATTAT TTTTGGAAAATTCAATGAGAAATCCG |
| Sequence-based reagent | Am_ex_20p3 | This paper | | TCTTGAGATCTTTCTGCATATAGCCGTGGTCTTCTTGCCCTCCCTCGCCCT CTGGCCCCGGACACCATCCACGGAGCTCCTTCCCTTCCCTTCACCGAATA TACTCGGCTGTGCAGCGCCTGCTACCCTCTCGCTCTACTCTCTTGCCTTT CCACCAGTCATGACAAGCCGGTCCGACTGGTACCCCCACCAACGCGGCC GGACGGACCCTTCTTGGGCCTCGCAAGGGCCCTGCGGGACCCCTCCCTC CTACATTCCAGCGGGGCCCCATCACGGCGAGGCTGAGCTGGCGGGTTTT GAGGCGCGCGAGCCATGCCACGACAGGAAAAAAATGCATCTGAAAAACGA AAACAAGTAGAAAAAGGTGGCTCACACCCCTGCATGCGTGCGTCGGTTTG CGTGAACGTTGCCCGGACCCCGTACCGAGGCCTCCTCCTCCTCCTCCTTT CTCCTCCTTTCTTCCTTCCTTTCTTTTCGCTCCTCTACCTCCTCTGCGCGCC TCTTTACGCTCCTCTTCTTGCTCTACGCTCTCGCGTACGTGCCCGCAAACTG CTGCCTGCCTGTTCAACGCTTCTTCTTCGTTTCCTTCTTCTTCTTCTTCTTCT TCCTCCTCTCTGCTTCGTCCTTGCGTTTCTCTCGATTCGCGTCACATCTCCG CTCCCCCAAATACGTTTCCTTTCTAAGATCGTTT |
| Sequence-based reagent | Am_klu_20p2 | This paper | | TTATTTATCGCCCTCGAAAGCGCTCGTCCTCTGCAGATTTCGATCGAGTCGT TCGACTTCGATATAAGAATTTCAGTGTAAACGCGATACACGTTAAATAACGAAT ATTTACGGACAAAGTCGGGCGAACGACGCGATCCTGGCCGCTCGTGGCCGA TGCGCAGGAAGGTAGGAGAGCGGAGAGGTTTTACGCTGTTCGCGGAGAGGA GGATAGGTTCGTATAGCTCCTTAAATCAACCCTAGTTGGCCTGTCAACCGAGT TGGCGCGCGCGCGCATTCCCTTTCGCGGCGCACAAATTACCACGCGTTTAAT TACCGTCCGATATACGAAGCAGGCTCATTAATCACCACGCCGATAACCCGTAAT TTTCCAGCAACGATAAAATCTATCGCGCGACACCGGCTCTCGCGACTTTCCTC TCTCTCTCTCTCCCTCTCTCTCTCTCGTTCGAGGAGAAAGGAGAAAAGGAAGC AGGAGGACGGAGGAGGTGTGCAGAGCGATCCTGTCGCCGCTTCCATATAGAT TTTTTTTCTCCCTTCCGCGCTCTTTCTCGCGCCAGTTCTCTTCGTGCGGCGGA AAATAGAGCGGCGCAACTCCCCTTCTCGCGACTCACGGAGGGCGAACAGCT GAAGCCGGCCGATCGATACGAA |
| Sequence-based reagent | Am_ush_20p8 | This paper | | GGGGCTCCCTCCTCCTTCCCTCCTCCTCCGTCTGTCCCAGTTGGTCAGCCA CGGTATCGTTTCGACGTCGATTCATCCCTCTTTTTCTCCCCCCTCTTCTCTCT CTCCTTCTGCCCCTCCCCCTCTCCCCTCCTCGCCATCGGCTTCGAGAGCCA CGAGGCGATCGAGAGAGAGAGAGAGAGAGAGAGAAAGGGTACCCCATCGA TGGATCGATCTATCGATCCACACGGGGATCCACCACGCTCTTCTGCCCTCTC CTCCTCCTCGCCACAATTTCTCTCCTCTTTACGTACGCTTCTCTCGTCCTTCG TGCCGCTCTTCGTCGCCATCGAGATTACGGCGAGCGAGGGGCCGACAGCCG AGGGGCTTCTTCCAATACTTTGTAAGTTTATTTGTATGATCCGCCAATACTTTGT ATCTTTATTTATATGAAATCGGATGGCGGATCGAGATTGCTCTCTCTCTCTCT CGTGTCGCTCGTGTCTCGTCTCGCTTCTCCCCCCGTTTCCTCTTAAAATTAATT ATACGTCCAAGGTGGGCGTAAGAGAGAGAGAGAGAGAGAAAGAAGTCGCAAT GAAACCGGAAGGATAAAGAGAATCCGATGGTGCGCACACGCACGTGTATGTAC ACGTCCACTTTATAACACTCG |
| Sequence-based reagent | Aa_hth_35p9 | This paper | | TTCCGAACACCTTGATTCAAATCCGAACAGTAGGTACGAATAAATCATACCGTTT TGCTTCGAAATCTGGACACCTAAGACGAAGTGTATTTTCAAACTTGAATATATATA TAGTAGAGATGGTCGGGTTTCACATTTTTCAAACCCGAACCCGACCCGTACCCG ACTTATTTTATTTCTTCGAACCCGGACCCGACCCGAACCCGAGACCCATAATTGAA AAGCAAACCCGGACCCGACCCGAACCCGAAAATTTTTCACAGTGCAAACCCGA ACCCGACCCGAAACCCGAAAAATGTTTGTAAAAAAACCCGAATACAACCCGAGT TTGAAAAGATGGTAAATTCATCGTTTCTGATGCATAAAGAAGCTTTTAGATTGTTA CTCTGTTCACAATTTTCACCAAACCCGACCTGAACCCGATTCAAACCCGACTTTT TGTAAGCCCGAACCCGACCCGTACCCGATAAAТТТCGTAGCCTACAAACCCGACC CGAACCCGAACCCGAAAAATTTCAAATATTCAAACCCGAACCCGACCCGAACCC GTCGGGTTCGGGTTCGGGTCGGGTTTCGCGTTTGAAAACCCGAGACCCGACCA TCTCTAATATATAGGCAAATTCATACATACTAAAAATCCAATGGTGATTCTTGCTTCG AAATCCGGACAGCATGTGAGAGCCGATTCAAATATTGGACACATTTGCTTCGAATT CCGGACACTTCTATTTATCTAGTTTGTTCAAAGATTC |
| Sequence-based reagent | Aa_ubx_26p0 | This paper | | AAGTCGGGTTTAGTCGGGTTTGTGTCGGGTTTGGTGAGAATTGTGAACAGAGTGA CAAACTAAAACCTTCCCTATGTATCAGAAACGATGAATTTATCATCTTTTAGAACTCG GGCTTTATTCGGGTTTTCCTTAGAAAACTTTTTCGGGTTTCGGGTCGGGTTTGGGT TTCAACAGCGAAAATTTTTCGGGTTCGGGTCGGGTCCGGGTTTGATTTTCAATTAT TGTCTCGGGTTCGGGTCGGGTCCGGGTTTGAAGAAATAAAATAAGTCGGGTACGG GTCGGGTTCGGGTTTGAAAAATGTGAAACCCGACCATCTCTACTCTTCAGGTAGTC GAGAGTTGTTTTTTTTTTATCTTTTATTTTTATTTTAAAGGCACTCTGTGCTCGTGCCC ACTACTATGCCGAAATCAGTTCATCTGTATCTTCTTCACCGATTAAGATCTATTTTTAA CTAATCTATATTTAAATCTACTTTCACTCTCTTCTACTCGTTTGCTCTCATACCGAGCA GGTAGGAGAGTGCTCTGCTGATAGTCCAATCGATTTCCATAAGCCATAGTTCCATTG CTCTTGCGGTGGTTCGTTTTGCCATGTTCCTGAGTCGTTTGAGGCTAGCTGCCTGC GAAGTGGGTCAGTTTGTCTCAG |

*Appendix 1 Continued on next page*

Appendix 1 Continued

| Reagent type (species) or resource | Designation | Source or reference | Identifiers | Additional information |
|---|---|---|---|---|
| Sequence-based reagent | Aa_psq_21p5 | This paper | | CCGAATTTGTGAGAGAGATAGGAGCCAATGTTTGAGTGATTCCCGCGAAGAATTGAAACCTATAAACGATTCCCACTAATTTTTGCAACATCTGTGATTTTTGATTTGATTTGAAACTGCAACTGACAGAAGATAATCAAAATACACTTTTTTCGCATTCGTACATCAATTGACAACCATCACTTGACACACCTGGCGATATGGACCAATAGGTCTGTGCCACAATAAGGGAGAGAAAAAAAAAAAAAAAGTGTGAAGCAAAAACACGCACATGTAACTTAAAGCACCACAAGAACCCTTTCAGCACCGGCCGCTTATGCTGATTTTATTAAAAAGCTTTATGCATACATGTACATAAGAGTGAGCATGCCGAAGCTCGAAAGTGTGTGTATGTGCGAATGCGCCAAAACACGATTATGTTTTCGTTTGTATTTCTTTCTTTTGCCGGCAAAAATTCTGTGTTTCGTTTTTTGATAGTAGGTAACTATGCCCACACAGTTACGGATCACACATAGTCATGGATCACTTTGGCGTTCAACATCGGATAACTCGCTCAAAACATATTTGCATGTGATGTAAACATATTTTTGCCAAGTCATAATATTTGTCTTCTGTCATTTGTAATATCAAATAATAAGCATAACATTAAACCGCAAAATAATGGTGTTTTTGAAAAATGTTTAGTTTGTATTGCCAAGCTATCATTAAATAGTCATTTATTGTAAGAAGTGCCATCAGCATTCTCCTATGCTTTTAGGTGATAAAATTCAAATATTATACATAATAGTTCCTCGTTCTCTTGAATTCAGTATGATTTCTTTGTTAGAAAACATTTTTCTTGTTTGTTGATACTGAATATTAGCAATTCCAACTAGTGATATTAGCAATTC |
| Sequence-based reagent | Tc_hth_15p5 | This paper | | TAATCTTTTAATTTAAAGCGTAGCTGAGCAGCTGGCTCTAATTCCACTTTCCTTATTTGGTTTCGTTGGTGTGGATTTTTGAAACGGATTATTTCGAGAAATAATAGTTTATTAGTGGTGGAAATAATGAATGGGTCTGGAGCGAGTTCCAGAGTGCGATTGGTTGGTTAGCGGGTAAATTTTTAAAAAGTGGGTGTCTTCTCCGACGGCAATTTAACGATCGTAACGACGTCGTCGCTAATTAGGCTCGTTGAGGCCGTCGCTAGATCGATAACACAGGCTGCGACATCGTCACAATGCACCGGTCGGGTTACACATCGGAGTCCGTCTCCCGGGGGCCCGTCTCAGATTCTCCGTATTAAAACACCGACATGTAAAAATATGGAAATTGCGCGCGGCAGAATGCGGTCCGATCAACCGGATGGCCATCGCGCATCGCTTTGCATTCGCAGCCGCATTTAAATTGCTAAAAGGGGACACTATCGAGCGGTCCATCTCTCTCGCAGCGTTGCGATATTATAATCTTGTTGCAAGGTAAATGCACATAACCGGTTACCCCAGACAGACGACGTCTTTGACACGAAAAAACCTGCCATCTATGTACAGCGGATCCTAATTTACGGCCTTATTCCATGTCATTAAGAGCATACGGGACGGACACGTTTTTAGGAACTTCGGACCCGACTTATCTCCGCGGACCGATAAGGAAATGTGCCTCTGGACACCTAACTTTGCCGACCAACAAAATCATAACGCTCGCTCTATGCCCATTGGGCAACACGAAAAAACCTGCC |
| Sequence-based reagent | Tc_Ubx_17p4 | This paper | | TGCATGTATGTCGAGTGGGTCCGGATGATGCGAACTCCCGCCGATTTCTTCGCAATCTGCAAATTCGCTCAAGTAGCTTAATAACAATGACAAAAGTGAGGCGGTATATTTCCGGCCGTCCGTTGAAAATTGTAATGATGTTATTAAAATTATGACGTGGCCGTGATGGTCGCCGAATTCTGGCGAAACGGCCGCGTAAAAACGGCACATAATTGGCTGACATTAAGATGTATCTGGAGATGTTTTTCGAATGCCTTCGTCCGGCGCGAATGCCTGAATAAGCGGCAAAGCTCGGAAAGCTCTTATAAATAAAAATGTACGGAGCCAATCAGATCGGCGAGTAAAAAGTACGTCTTTTCTTACACCAGAGGATCGCAGCTGCCGCAGAATCCGGTCGCGGATAAGAAATAAGAAGTGCTGCATAAATGCATTGATCATTCGCCGGGTCTCCGTCTGCTGTTCCTCAGCGAGAAAACGGGTTTAAGTCTGGATACTTTTGGCTCTCTGGAAAGTGCTTTTTGCATTAAGCTGCCGAGAGAGAATAAAGACGTTTGCGGTGTCGGACGGTGACCAATGCTGCTGCTGCTGCTCTGCCTTCCAAGTGCGTGCTTTAAATCTTCCACTTTGCAAGTAAATCGAGACGAACGCTGAATATTTTACACGAACACTGTTTATAGCCCAAATAACAGCCTTCCAAGGGCGGCCACCATCAAAAAATGGAGCGCTCAAACCCGAAATATGGGCGGGCGAAAATTATTCAAACCACAAAGCGAGGAAATCAGAAATTCAAAAATTGACGGCTTTCAACTCAGGACTGAATTTTTATAAATTTTTGTTCGCTACTGCAATTTGGGACAGAAAATTACATCT |
| Sequence-based reagent | Tc_Ubx_19p9 | This paper | | CGTATTTAAATATCGTTAGGTTCGATGGTAAAATTGGAGAAAATTGTCGCGCGCGTTTAAGACAAAGAAAATTCCCGTCGGGTTATCAATCTTGGGTTATCTGTACCCTCGGGCCGAAAAACTCTGTAAAGAAGAGACAAAAGGACGTGACAGTCCAATTTCCATTTCAGATCGAAATTGTTCGCCCCCCGGAAGTTTATCGGGGCCCGTTGGCGGAATTAATAAATTGGTGCGCGACTTAATTGCGGCGATAAAGAAGAGAAGAACACGAATGAGGGACGGCGACAAAAATATTATTTGCTCGTGAACGAGGAGGCAAAGGGCATTGATATCTCGTGCAACGCCGGATATTGGCTGCTTCTGGTCGCGGTTTGCGGGGCTTCTAAGACTGTGCAGGGTTTGGGGAGCGGCCCCGAGCTCGAGAGAAATTATGTACGAGGCATTGGGAGCAATATATCTCCGGCCGGGACGTGCCAGACAGAGTAGACGGGGTATTATATAGGAAGGAAGGAACCTGAGGCCGGGGCCGGAGCCTCCTCGTCCCCAGGCGCTCGTCCCCCAGAATGAGACACTTGCCGCCAAGTCCACCGCCTTAAATTGTCATCTGAAGAAAGAAACTTCATTACGAACTACGCCCTCATTTCTTTGCGAGGCGATCCATCGCGCAAAAGCAACGCACGCATTTTGCAACAACTATTCAACCACTAAAATTAAACGAATTTCAAACCTATTCCGGATTAATGATTTCCTCCTCGATTCAAGCTAATTGGGTGTTTCCTAG |

*Appendix 1 Continued on next page*

*Appendix 1 Continued*

| Reagent type (species) or resource | Designation | Source or reference | Identifiers | Additional information |
|---|---|---|---|---|
| Sequence-based reagent | Tc_psq_19p7 | This paper | | GGATTTTTTAGATAGATCATCAAGTTAAAAGTGCTTCGAATATATGTCATCAAAAATA AGATCAACTGATGGCTTTCTTTGCTTTATTCCCAATCTACTGTTAGAAAATCAACAA CAACTAAGTTTTCTGTAAAATATAGTTCTTTCGGTGGCAAGAATAATATTATAATCGG GTTTCTTCTGCCTTATATTCTGTTTTCTTTGCTCCTATGTTAGTGCAAGTGTGTAAC TTGGCGAACTCTTTCGAATTATCAAGGAAGTGTGAGTTTTATGAGAAAACAGCTAA AGTCGCCCCTAATTTGTTGACTTATTTGCTTTCGTTGGTTCTCCCGTTCTTTGGAG TATGTCGTCCGGTTTTTCTTATGAGCCATAATTACAAATTTCCATTTTCGGTTTTCG GCTCGCGTTCGTTTTGGAAAAGAGCGAATGTGCGGCGCGTTCATTTTCAATTTTG CGCGACCGTCCGACATTTTCCAATTTTCCGTGCAAGGACGAGGAGCGAGTGCAA AAAATGGCAGTCCTTGTCTGCAAAAAGCCCCAATTAAAACCGAAGTTGTAGTAGT GCGTGCGCCGAGCATTTCTCTCGATCTATCACGGGGTAGCAGCATCCCTCCGTA GGCTCACTCTCTGGCCAGTCTTAGTTTGCGCTTTCCCCGGAATTCACTGAAGGT CGTCGAGGTCGCAAGTAAGTACACAGTGCATGTGCACTTGCATGCATGCTTGCAC TTTCTGTGCCCCCGCGCGCCGCCGCCGCCGCCGCCGCATTAGCGTCTCTGTTTTGGTC CTTATATCCATCCGCTGTTCCCTTCTTCTGTCTATCCTTCAACTTCCTTCGCCGCTCG CCAGCTCCGGGACGCCACTCCATCATAAAACTGCGACCGCAAAAGCGACACTCATT ATCGATTGCTCCAAGACGAATTAAAAGCCGCAGCGCTCCCCAAAACCGGGTTATTTT TTTCGGAATTTTGCTGGCTTGGAGCGGACTCCCAGACGATCCCCGGACTAATCCGG AGGGTTGCCTGGCGAGCGGCATTCGGCTTTAGGCTCCGGGGCACGCATTGGGGGA AAGTGATGCGGTGTCTGGACCAATCAATACCGGGTTCAAGGACGGCTTCTCTTATAT GTGTATGTGAGCTTCCTTTTCCCGCTCGTCAAAACGGGACAAGACGGGAATTAATTG CACGACAATTGGGACGCCGACTCCACAGATGGGGCGACAAAATGGACGCAACGAA CTAAATCTATTCACTTTT |
| Sequence-based reagent | Am_Ubx_0p39 | This paper | | TCTCTCTCTCCTCGAGTGTAGCATATATCCATTCCACCATCGATCGAGGATTTCGATC CCCCTTGGACTCATGCTGCGATATTCGATCGTCCCTCCCCCCACTCCTCCGCGCTC TCATTCGATCCTTCTTTTCTTCCCTCCCCCCCAACCACTTTGATCCTTCTCTCTCTCT CTCTTCCTTCTTTTTACTTCTTCTTCTTCTTGCTGCTGCAACTACCCGCTGCCTCTAA CCGCTAGCCGGACAAAACATTTCTTAATTGGGTTTCGTTCGGAAAGAACCGTCCGAT TTCGTTTCGCAAGGGATCCAGCCTGCTGCTTCTGCCGGTTTTACCGCGTCTCTACGT GGCTTCGTCGTTCCCTCCTCGTTCTGCTTGCCTTTCCTTCGAACGATTATTTATTTCG TCGTTCGAATTCCTTATTTTTCCATCCTGTTATCCCTTATTGTAATAAAGTAAAAATAATT GAATTTTCCTTCGAACGAGCGAAGTTTGTTCTAATC |
| Sequence-based reagent | Am_Ubx_37p2 | This paper | | AGAGAGAGAGAGAGAGAGAGAAAGTCAGGCAGACGGAGACAGAGGAATGGGTTGGGT AAGGGGGATAGAGTAGGGGCGGGAGGGCGTTCCACGGCACCCTGCATGGGGTAGC TTGCAACCTCACGCGACACTAGAGCCATCTATATCCCCGGAGATTTATGAGTTCCTGG TGCAGCGGCTGCTCGCAGCAACTACACACCACGCAGTATCGGGTCCGGTGTTGGTG CTGCCCCCTGTCGCGACGGGCGTGCTGTTGCCCCGCGGGGGTTACGCGCAATTCG CGCTCCGTGCAACGTCGCCCTGATAAAAAACTCTTGCGACTCGATCTCAATCCCGAT GCTTCTGCGAACCTTCCCTCCGTTCTCGCGCCCGTCTCGTCGTGCGCCCGTCGCGGT CGTCTCTCTCTCTCTCGCTCTCCGGTGTTGGCGGGCTATCGGATCTTCTCTCTCTC TCTCGCTCACTTGGTTCGCTTCTTGCTTTCGATGCGACGCGACGACCAGCGATCTCAC TTTCTCTCTCGCTCTCGCTTTCGAGCTCGCACTTGAAATATCGATCATCATTGTGTTTCC TACGCATTTGTAGACCGCAAACGCGAAATTATTATGGGCCTGTGCACGTTTGAATTTCTT ATATTCTTTTTTTTCCATTATACGCTAGGTTAGCGTAGATATAATTCTGCTAAATATAGTGAA GATAATTCGAATTAAATTAAATTGAAAATTTTCGTATTACATAATACTGTTTCGTTATTTATA ACTGATTAGAATATTTATTGATACCAAATGAAATTTTTGGTAAACTCTCGAACATTGTTTCAT TCTTCTATATCGTATTGGTGAAAAATTACATCTCGATTTTTTTCTCACGAACTTATATCGCGGT AAAAGAACTGTGGACAACTGTGCAGCATCTCCTCGCTCGATGAAGTCATTTGAACGAGCA TTCCTCGGCCGATCTCAGATACAATCTCCTTCAAACAAAGAGCTCCATTGCCGCGTGCA |
| Sequence-based reagent | Am_psq_29p2 | This paper | | TCGCACGAGTACATAACGCTACCTTTGTCGCGTCGAAGGTAGAGGCACGATTCTGTCC TTTCCCGTTCTCTCGCGAACCTTGCATCCGTCTTCGTCTCGCTGTGGCCAAACGCGTG CTAGGTCTTCGTCTTCCACATTCCGTCTCGTTCGTTTCCGCACAGACTATATTTCTGTTC TCGTTTAGCCGCGGAAAGTCTTGCTCGCTCCCACGGGAACCACTCGTCGATGCTCGTC GCTTAACCGTCAGAGGCGAGCGCGCATTTCTCTCAAACACCGCAGACTTGCCTCTCCG CCGATCCCCGTTCCCACCCCCCGGTGCTCGATGCTCTCTGTCACCCCTCCACCAAACGG ACTCCTACCGGCCCGCTCCCCTCGCTTTGCGCCGCTTTCCACCAACCGTCCTGCCACC CGCCGGTTTTCAACCCCTTTCCCCGCTCTCTCGGCGACTGGTCAGGTGCGCTCGCTCG CTCGCTCGCTCCACGCGTACGCTCAATCGCTCTCTGTCCACCGCCGAGCACGCATCCC CCGCGAGTCTCTTCCTCGTTGTACGCGCTCGAGCGCGGATTCAATCCGTCCTTGTTCGT CGCGTCGGCGAATTTCGCGGCGTCCTCCGCCGCCGCCGCCGCCGCCGCCGCCACCTCTTC CTCCTCCTCCTCCGCCTCCTCCTCCTGCTGATACTCCTCTTCCTCCTCGGT |
| Software, algorithm | SCRMshaw pipeline | **Asma et al., 2024**, doi: 10.17504/protocols.io. e6nvw1129lmk/v2 | | |
| Other | REDfly database | **Keränen et al., 2022**, PMID:35886794 | RRID:SCR_006790SCR_006790 | Database with results information, see "Results—An insect regulatory annotation resource" |

