## [Editor Report · eLife assessment]

In the revised version of this **important** study, the authors present a **convincing** pipeline for insect genome regulatory annotation across 33 insect genomes spanning 5 orders. Despite technical limitations in the field owing to the lack of comprehensive knowledge of enhancer content in any system, the authors employ several independent downstream analyses to support the validity of their enhancer predictions for a subset of these genomes. Taken together, the revised results suggest that this prediction pipeline may have uses in identifying functional enhancers across large phylogenetic distances. Reviewers note caveats that an experimental validation is not yet available in the field to validate a large class of newly identified enhancers across such evolutionary distances, and other pipelines might be of use to compare. This work will be of interest to the computational genomics, evolutionary biology, and gene regulation fields.

---

## [Referee Report · Reviewer #1 (Public review)]

Summary:

The authors provide an genome annotation resource of 33 insects using a motif-blind prediction methods for tissue-specific cis-regulatory modules. This is a welcome addition that may facilitate further research in new laboratory systems, and the approach seem to be relatively accurate, although it should be combined with other sources of evidence to be practical.

Strengths:

The paper clearly presents the resource, including the testing of candidate enhancers identified from various insects in *Drosophila*. This cross-species analysis, and the inherent suggestion that training datasets generated in flies can predict a cis-regulatory activity in distant insects, is interesting. While I can not be sure this approach will prevail in the future, for example with approaches that leverage the prediction of TF binding motifs, the SCRMShaw tool is certainly useful and worth of consideration for the large community of genome scientists working on insects.

Weaknesses from the previous version were appropriately corrected in this revision, as the authors improved data availability including with genome annotation resources.

---

## [Referee Report · Reviewer #3 (Public review)]

Summary:

In this ambitious paper, the authors develop an unparalleled community resource of insect genome regulatory annotations spanning five insect orders. They employ their previously-developed SCRMshaw method for computational cross-species enhancer prediction, drawing on available training datasets of validated enhancer sequence and expression from *Drosophila melanogaster*, which had been previously shown to perform well across select holometabolous insects (representing 160-345MY divergence). In this work they expand regulatory sequence annotation to 33 insect genomes spanning Holometabola and Hemiptera, which is even more distantly related to the fly model. They perform multiple downstream analyses of sets of predicted enhancers to assess the true-positive rate of predictions; the independent comparisons of real predictions with simulated predictions and with chromatin accessibility data, as well as the functional validation through reporter gene analysis strengthen their conclusions that their annotation pipeline achieves a high true-positive rate and can be used across long divergence times to computationally annotate regulatory genome regions, an ability that has been largely inaccessible for non-model insects and now is possible across the many newly-sequenced insect scaffold-level genomes.

Strengths:

This work fills a large gap in current methods and resources for predicting regulatory regions of the genome, a task that has long lagged behind that of coding region prediction and analysis.

Despite technical constraints in working outside of well-developed model insect systems, the authors creatively draw on existing resources to scaffold a pipeline and independently assess likelihood of prediction validity.

The established database will be a welcome community resource in its current state, and even more so as the authors continue to expand their annotations to more insect genomes as they indicate. Their available analysis pipeline itself will be useful to the community as well for research groups that may want to undertake their own regulatory genome annotation.

Weaknesses:

The work here is limited by the field-wide lack of an independently validated set of tissue specific enhancers that could be used to directly benchmark this pipeline. The prediction of true positive enhancer identification rates and in vivo reporter gene assays offer some insight into the rates of successful prediction, but the output of SCRMshaw regulatory annotation should be regarded as another prediction-generating tool.

---

## [Author Response]

The following is the authors’ response to the original reviews.

**Public Reviews:**

**Reviewer #1 (Public Review):**
Strengths:The paper clearly presents the resource, including the testing of candidate enhancers identified from various insects in *Drosophila*. This cross-species analysis, and the inherent suggestion that training datasets generated in flies can predict a cis-regulatory activity in distant insects, is interesting. While I can not be sure this approach will prevail in the future, for example with approaches that leverage the prediction of TF binding motifs, the SCRMShaw tool is certainly useful and worth consideration for the large community of genome scientists working on insects.

We thank the reviewer for the positive comments, and would just like to point out that we agree: while we cannot of course know if other methods will overtake SCRMshaw for enhancer prediction—we assume they will, at some point (although motif-based approaches have not fared as well in the past)—for now, SCRMshaw provides strong performance and is a useful part of the current toolkit.

Weaknesses:While the authors made the effort to provide access to the SCRMShaw annotations via the RedFly database, the usefulness of this resource is somewhat limited at the moment. First, it is possible to generate tables of annotated elements with coordinates, but it would be more useful to allow downloads of the 33 genome annotations in GFF (or equivalent) format, with SCRMshaw predictions appearing as a new feature. Also, I should note that unlike most species some annotations seem to have issues in the current RedFly implementation. For example, Vcar and Jcoen turn empty.

We have addressed these weaknesses in several ways:

(1) We have created GFF versions of the SCRMshaw predictions and provide them standalone and also merged into the available annotation GFFs for each of the 33 species

(2) We have made these GFF files, and also the original SCRMshaw output files, available for download in a Dryad repository linked to the publication (https://doi.org/10.5061/dryad.3j9kd51t0).

(3) We have added the inadvertently omitted species to the REDfly/SCRMshaw database.

We agree that the database functions are still somewhat limited, but note that database development is ongoing and we expect functionality to increase over time. In the meantime, the Dryad repository ensures that all results reported in this paper are directly available.

**Reviewer #2 (Public Review):**
Summary:… Upon identification of predicted enhancer regions, the authors perform post-processing step filtering and identify the most likely predicted enhancer candidates based on the proximity of an orthologous target gene. …

We respectfully point out a small misunderstanding here on the part of the reviewer. We stress that putative target gene assignments and identities have no impact at all on our prediction of regulatory sequences, i.e., they are not “based on the proximity of an orthologous target gene.” Predictions are solely based on sequence-dependent SCRMshaw scores, with no regard to the nature or identities of nearby annotated features. Putative target genes are mapped to *Drosophila* orthologs purely as a convenience to aid in interpreting and prioritizing the predicted regulatory elements. We have added language on page 8 (lines 189ff) to make this more clear in the text.

Weaknesses:This work provides predicted enhancer annotations across many insect species, with reporter gene analysis being conducted on selected regions to test the predictions. However, the code for the SCRMshaw analysis pipeline used in this work is not made available, making reproducibility of this work difficult. Additionally, while the authors claim the predicted enhancers are available within the REDfly database, the predicted enhancer coordinates are currently not downloadable as Supplementary Material or from a linked resource.

We have placed all the code for this paper into a GitHub repository “Asma_etal_2024_eLife” (https://github.com/HalfonLab/Asma_etal_2024_eLife) to address this concern. As described in our response to Reviewer 1, above, all results are now available in multiple formats in a linked Dryad repository in addition to the REDfly/SCRMshaw database.

The authors do not validate or benchmark the application of SCRMshaw against other published methods, nor do they seek to apply SCRMshaw under a variety of conditions to confirm the robustness of the returned predicted enhancers across species. Since SCRMshaw relies on an established k-mer enrichment of the training loci, its performance is presumably highly sensitive to the selection of training regions as well as the statistical power of the given k-mer counts. The authors do not justify their selection of training regions by which they perform predictions.

Our objective in this study was not to provide proof-of-principle for the SCRMshaw method, as we have established the efficacy of the approach at this point in several previous publications. Rather, the objective here was to make use of SCRMshaw to provide an annotation resource for insect regulatory genomics. Note that the training regions we used here are the same as those we have used in earlier work. Naturally, we performed various assessments to establish that the method was working here, but we make no claims in this work about SCRMshaw’s relative efficiency compared to other methods. Some of our prior publications include assessments of the sort the reviewer references, which suggest that SCRMshaw is at least comparable to other enhancer discovery approaches. We note that benchmarking of such methods is in fact extremely complicated due to the fact that there are no established true positive/true negative data sets against which to benchmark (we have explored this in Asma et al. 2019 BMC Bioinformatics).

While there is an attempt made to report and validate the annotated predicted enhancers using previously published data and tools, the validation lacks the depth to conclude with confidence that the predicted set of regions across each species is of high quality. In vivo, reporter assays were conducted to anecdotally confirm the validity of a few selected regions experimentally, but even these results are difficult to interpret. There is no large-scale attempt to assess the conservation of enhancer function across all annotated species.

We respectfully disagree that there is insufficient validation. We bring several different lines of evidence to bear suggesting that our results fall into the accuracy range—roughly 75%—established both here and in previous work. We are also clear about the fact that these are predictions only and need to be viewed as such (e.g. line 638). Although “large-scale” in vivo validation assays would certainly be both interesting and worthwhile, the necessary resources for such an assessment places it beyond our present capability.

Lastly, it is suggested that predicted regions are derived from the shared presence of sequence features such as transcription factor binding motifs, detected through k-mer enrichment via SCRMshaw. This assumption has not been examined, although there are public motif discovery tools that would be appropriate to discover whether SCRMshaw is assigning predicted regions based on previously understood motif grammar, or due to other sequence patterns captured by k-mer count distributions. Understanding the sequence-derived nature of what drives predictions is within the scope of this work and would boost confidence in the predicted enhancers, even if it is limited to a few training examples for the sake of clarity of interpretation.

Again, we respectfully disagree that “this assumption has not been examined.” Although we did not undertake this analysis here, we have in the past, where we have shown that known TFBS motifs can be recovered from sets of SCRMshaw predictions (e.g., Kazemian et al. 2014 Genome Biology and Evolution). We return to this point when we address the Comments to Authors, below.

**Reviewer #3 (Public Review):**
Weaknesses:The rates of predicted true positive enhancer identification vary widely across the genomes included here based on the simulations and comparison to datasets of accessible chromatin in a manner that doesn't map neatly onto phylogenetic distance. At this point, it is unclear why these patterns may arise, although this may become more clear as regulatory annotation is undertaken for more genomes.

We agree that we do not see clear patterns with respect to phylogenetic distance in our results. However, we note that this initial data set is still fairly small, and not carefully phylogenetically distributed. We are hoping that, as the reviewer suggests, some of these questions become more clear as we add more genomes to our analysis. Fortunately, the list of available genomes with chromosome-level assembly is growing rapidly, and as we move ahead we should have much greater ability to choose informative species.

Functional assessment of predicted enhancers was performed through reporter gene assays primarily in *Drosophila melanogaster* imaginal discs, a system amenable to transgenics. Unfortunately, this mode of canonical imaginal disc development is only representative of a subset of all holometabolous insects; therefore, it is difficult to interpret reporter gene expression in a fly imaginal disc as evidence of a true positive enhancer that would be active in its native species whose adult appendages develop differently through the larval stage (for example, Coleopteran and Lepidopteran legs). However, the reporter gene assays from other tissues do offer strong evidence of true positive enhancer detection, and constraints on transgenic experiments in other systems mean that this approach is the best available.

Please see an extensive discussion of this point in our response to Reviewer 3, below.

**Recommendations for the authors:**

**Reviewer #2 (Recommendations For The Authors):**
Major Concerns:(1) While the GitHub source code for SCRMshaw is provided, the authors do not provide a repository of manuscriptspecific code and scripts for readers. This is a barrier to reproducibility and the code used to perform the analysis should be made available. Additionally, links to available scripts do not work, see Line 690. Post-processing scripts point to a general lab folder, but again, no specific analysis or code is sourced for the work in this specific manuscript (e.g. Line 637).

As noted above, we have corrected this oversight and established a specific GitHub repository for this manuscript “Asma_etal_2024_eLife” (https://github.com/HalfonLab/Asma_etal_2024_eLife).

(2) On lines 479-488, there is a discussion about the annotations being provided on REDfly, though no link is provided.

We have included a link in the text at this point (now line 515).

Additionally, for transparency, it would be valuable to provide in Supplementary Table 1 the genomic coordinates of the original training sets in addition to their identity.

These coordinates have been added to Supplementary Table 1 as suggested.

Also, it is suggested to provide genomic coordinates of the predicted enhancers for each training set across all species, perhaps with a column denoting a linked ID of one genomic coordinate in a species to another species (i.e. if there is a linked region found from *D. melanogaster* to J. coenia, labeling this column in both coordinate sets as blastoderm.mapping1_region1). Providing these annotations directly in the work enhances the transparency of the results.

We are unsure exactly what the reviewer means here by “a linked region.” It is critical to understanding our approach to recognize that the genome sequences have diverged to the point where there is no alignment of non-coding regions possible. Thus there is no way to directly “link” coordinates of a predicted enhancer from one species to those of a predicted enhancer in another species. The coordinates for each prediction are available on a per-species basis either through the database or in the files now available in the linked Dryad repository; these can be filtered for results from a specific training set. The database will allow users to select all results for a given orthologous locus, from any subset of species. More complex searches will continue to become available as we improve functionality of the database, an ongoing project in collaboration with the REDfly team.

(3) Figure 2B: It is unclear what this figure shows. Are the No Fly Orthologs false positives, Orthology pipeline issues, or interesting biology?

We have clarified this in the Figure 2 legend. “No Mapped Fly Orthologs” indicates that our orthology mapping pipeline did not identify clear *D. melanogaster* orthologs. For any given gene, this could reflect either a true lack of a respective ortholog, or failure of our procedure to accurately identify an existing ortholog.

(4) SCRMshaw appears to be a versatile tool, previously published in a variety of works. However, in this manuscript, there is little discussion of the sensitivity of SCRMshaw to different initial parameters, how the selection of training loci can impact outcomes, or how SCRMshaw k-mer discovery methods compare to other similar tools.- This paper would be strengthened by addressing this weakness. Some specific suggestions below:In order to strengthen confidence that SCRMshaw is a reliable predictor of enhancer regions in other species, it is suggested that you benchmark against other k-mer-derived methods to assign enhancers, such as GSK-SVM developed by the Beer Lab in 2016 (https://www.beerlab.org/gkmsvm/, https://www.biorxiv.org/content/10.1101/2023.10.06.561128v1).

We have established the effectiveness of SCRMshaw as an enhancer discovery method in previous work, and the main goal of this study was to make use of the established method to annotate numerous insect genomes as a community resource. Our claim here is that SCRMshaw works well for this purpose; we do not attempt a strong claim about whether other approaches may work equally well or marginally better (although we do not believe this is the case, based on prior work). Benchmarking enhancer discovery is challenging, as we point out in Asma et al. 2019 (BMC Bioinformatics), and, while important, best left for a dedicated comprehensive study. A major problem is that there are no independent objective “truth” sets for enhancers from the various species we interrogate here. Thus, while we could also run, e.g., GSK-SVM, what criteria would we use to establish which method had better accuracy for a given species? Note that the work from Beer’s lab took advantage of the ability to match human-mouse orthologous (or syntenic) regions and available open-chromatin data to assess whether conserved enhancers were discovered, but this is not possible given the degree of divergence, limited synteny, and relative lack of additional data for the insect genomes we are annotating.

- In Table S1, we see that 7-146 regions are used as training sets, which is a huge variety. Does an increase in training set size provide a greater "rate of return" for predicted regions? Is the opposite true? Addressing this question would allow readers to understand if they wish to use SCRMshaw, a reasonable scope for their own training region selections.- Within a training set, does subsampling provide the same outcomes in terms of prediction rates? There is no exploration of how "brittle" the training sets are, and whether the generalized k-mer count distributions that are established in a training set are consistent across randomly selected subgroups. Performing this analysis would raise confidence in the method applied and the resulting annotations.

These are interesting and important questions, but again we feel they are beyond the scope of this particular study, which is focused primarily on using SCRMshaw and not on optimizing various search parameters. That said, this is of course something we have investigated, although as with other aspects of enhancer discovery, the absence of a true gold standard enhancer set makes evaluation difficult. We have not found a clear correlation between training set size and performance beyond the very general finding that performance appears to be best when training set size is moderate, e.g. 20-40 initial enhancers. We suspect that larger training sets often contain too many members that don’t fit the core regulatory model and thus add noise, whereas sets that are too small may not contain enough signal for best performance (although small sets can still be useful, especially if used in an iterative cycle; see Weinstein et al. 2023 PLoS Genetics). However, establishing this rigorously is highly challenging given the limitations with assessing true and false positive rates at scale.

(5) In Figure 2C, when plotting hexMCD, IMM, pacRC, and then the merged set, it is unclear whether the scorespecific bar allows coordinate redundancy, though this is implied. What might be more useful is a revision of this plot where the hexMCD/IMM/pac-RC-specific loci are plotted, with the merged set alongside as is currently reported. This would give the reader a clearer understanding of the variability between these scoring methods and why this variability occurs.

We have added the breakdowns between IMM, hexMCD, and pacRC in Supplementary Table S2, and made more complete reference to this in the text (lines 682ff). Both the database and the data files in the Dryad repository allow exploration of the overlap between the different methods and contain both separate and merged (for overlap and redundancy) results.

Additionally, there is no information in the Methods section of these three SCRMshaw scores and what they represent, even colloquially. While SCRMshaw has been applied in several papers previously, it would help with scientific clarity to describe in a sentence or two what each score is meant to represent and why one is different from another.

We had chosen to err on the side of brevity given prior publication of the SCRMshaw methodology, but we recognize now that we went too far in that direction. We have added more complete descriptions of the methods in both the Results (lines 164-167) and the Methods (lines 667-681) sections.

(6) When describing results in Figure 2, an important question arises: "Is there an anti-correlation between the number of predicted regions and evolutionary distance?" This would be an expected result that could complement Figure 4's point that shared orthology across 16 species is rarer than across 10 species. Visualizing and adding this to Figure 2 or Figure 4 would be a powerful statement that would boost confidence in the returned predicted enhancers and/or orthologous regions.

This is an important question and one in which we are very interested. Unfortunately, we do not have sufficient data at this time to address this proper statistical rigor. As we remarked above in response to Reviewer 3, “We agree that we do not see clear patterns with respect to phylogenetic distance in our results. However, we note that this initial data set is still fairly small, and not carefully phylogenetically distributed. We are hoping that, as the reviewer suggests, some of these questions become more clear as we add more genomes to our analysis. Fortunately, the list of available genomes with chromosome-level assembly is growing rapidly, and as we move ahead we should have much greater ability to choose informative species.”

(7) In Figure 3, the authors seek to convey that SCRMshaw predicts enhancer regions that are mapped nearby one another, across different loci widths, and that this occurrence of nearby predicted regions occurs more than a randomly selected control. This is presumably meant to validate that SCRMshaw is not providing predictions with low specificity, but rather to highlight the possibility that SCRMshaw is identifying groups of shadow enhancers. However, these plots are extremely difficult to decipher and do not strongly support the claims due to the low resolution and difficult interpretability of the boxplot interquartile distributions.

Additionally, as the majority of predicted regions are around ~750bp, how does that address loci groups of <1000bp? This suggests that predicted regions are overlapping, and therefore cannot be meaningfully interpreted as shadow enhancers. This plot should either be moved to the supplements or reworked to more effectively convey the point that "SCRMshaw is detecting predicted regions that are proximal to one another and that this proximity is not due to chance".

- A suggestion to rework this plot is to change this instead to a bar plot, where the y-axis instead represents "number of predictions with at least 2 predicted regions proximal to one another" divided by "total number of predictions", separating bar color by simulated/observed values. The x-axis grouping can remain the same. Because this plot is a broad generalization of the statement you're trying to make above, knowing whether a few loci have 2 versus 4 proximal predicted enhancers doesn't enhance your point.

We agree with the reviewer that these are not the clearest plots, and thank them for the suggestions regarding revision. We tried many variations on visualizing these complex data, including those suggested by the reviewer, and have concluded that despite their weaknesses, these plots are still the best visualization. The main problem is that the observed data cluster heavily around zero, so that the box plots are very squat and mainly only the outlier large values are observed. The key point, however, is that the expected values almost never give values much greater than one, so that the observed outlier points are the only points seen in the upper ranges of the y-axis. This is true across the three species, across the bins of locus sizes, and across training sets (averaged into the box plots). The reviewer is correct as well about the bins where locus size is < 1000. However, inspection of the data shows that this is not a large concern, as very few data points lie in this range and we never see multiple predicted enhancers there. Thus we believe while not the prettiest of graphs, Figure 3 does effectively support the claims made in the text. In keeping with our view that it is preferable to have data in the main paper whenever possible, we choose to keep the figure in place rather than move it to the Supplement.

- Label the species for the reader's understanding of each subplot on the plot.

We apologize for this oversight and have now labeled each plot with its relevant species.

(8) SCRMshaw operates on k-mer count distributions compared to a genomic background across different species, allowing it to assign predicted regions without prior knowledge of an organism's cis-regulatory sequences. This is powerful and boosts the versatility of the method. However, understanding the cis-regulatory origins of the kinds of kmers that are driving the detection of orthologous regions across species is crucial and absolutely within the scope of the paper, particularly for the justification of the provided annotations. Is SCRMshaw making use of enriched motifs within the training region set to assign regions in other species? One would presume so, but it is necessary to show this. There are many motif discovery tools that are readily available and require little up-front knowledge and little to no use of a CLI, such as MEMESuite (https://meme-suite.org/meme/tools/meme). It is highly recommended that, even for a few training pairs that are well understood (e.g. mesoderm.mapping1, dorsal_ectoderm.mapping1), assess the motif enrichment within the original sequence set, then see whether motif enrichments are reflected in the predicted enhancers. As evolutionary distance increases between *D. melanogaster* and the species of interest, is the assignment of enriched motifs more sparse? Is there a loss of a key motif? These are the kinds of questions that will allow readers to understand how these annotations are assigned as well as boost confidence in their usage.

This is a very important point and a subject of significant interest to us. We have demonstrated in earlier work (e.g., Kazemian et al. 2014 Genome Biol. Evol.) that SCRMshaw-predicted enhancers do contain expected TFBS motifs, across multiple species—and that even an overall arrangement of sites is sometimes conserved. Thus we have previously answered, in part, the reviewer’s question.

What we also learned from our previous work is that filtering out relevant motifs from the noise inherent in motif-finding is both arduous and challenging. As the reviewer is no doubt aware, while using motif discovery tools is simple, interpreting the output is much less so. In response to the reviewer’s comments, we revisited this issue with data from a small sample of training sets. We can discover motifs; we can see that the motif profiles are different between different training sets; and we can observe the presence of expected motifs based on the activity profile of the enhancers (e.g., Single-minded binding sites in our mesectoderm/midline training and result data). However, to do this cleanly and with appropriate statistical rigor is beyond what we feel would be practical for this paper. We hope to return to this important question in the future when we have a larger and phylogenetically more evenly-distributed set of species, and the time and resources to address it appropriately.

(9) Figures 5-7 need to have better descriptions.

We have added to the figure 6 and 7 legends in response to this comment; please note as well that there is substantial detail provided in the text. If there are specific aspects of the figures that are not clear or which lack sufficient description, we are happy to make additional changes.

Minor Concerns(1) In Figure 1A, it is implied that "k-mer count distributions" are actually only "5-mer count distributions". However, in the published documentation of SCRMshaw, it is suggested that k-mers between 1-6 bp are involved in establishing sequence distributions. Please add a justification for the selection of these criteria. It would be helpful to understand the implications of using up to a 3-mer versus a 12-mer when assessing k-mer counts using SCRMshaw.

We have clarified in the Figure 1 legend that this is just an example, and the k-mers of different sizes are used in the IMM method; we have also increased the description of the basic method in the Methods section. To be clear, the hexMCD sub-method is 6-mer based (5th-order Markov chain), as is pacRC, while the IMM method considers Markov chains of orders 0-5.

(2) Control the y-axis to remove white space from Figure 2D.

We have amended the figure as suggested.

Additionally, expand in the manuscript on expected results from SCRMshaw. Given training regions of 750 bp, is the expectation that you return predicted enhancers of the same length? This is not explicitly stated, only a description of outliers.

The scoring is not dependent on the length of the training sequences, and there is no direct expectation of predicted enhancer length. Scores are calculated on 10-bp intervals, and a peak-calling algorithm is used to determine the endpoints of each prediction based on where the scores drop below a cutoff value. Thus there is no explicit minimum prediction length beyond the smallest possible length of 10-bp. That said, the initial scoring takes place over a 500-bp sequence window (for reasons of computational efficiency), which does influence scores away from the smaller end of the possible range. We correct for this in part by reducing scores below a certain threshold to zero, to prevent multiple low-scoring regions from combining to give a low but positive score over a long interval. Indeed, we found that in the original version of SCRMshawHD (Asma et al. 2019), multiple low-scoring but above-threshold intervals would get concatenated together in broad peaks, leading to an unrealistically large average prediction length. In the version used here, described in Supplementary Figure S6, low-scoring windows are now first reset to zero and a new threshold is calculated before overlapping scores are summed. This helps to prevent the broad peak problem, and we find that it results in a median prediction length ~750 bp, more in line with expected enhancer sizes.

**Reviewer #3 (Recommendations For The Authors):**
Line 161: Given that the SCRMshaw HD method is the basis for the pipeline, the methodology deserves at least an "in brief" recapitulation in this manuscript.

As we remark in our response to Reviewer 2, above, “We had chosen to err on the side of brevity given prior publication of the SCRMshaw methodology, but we recognize now that we went too far in that direction. We have added more complete descriptions of the methods in both the Results (lines 164-167) and the Methods (lines 667-681) sections.”

Line 219: Throughout the reporting of the results, there appeared to be a bit of inconsistency/potential typos regarding whether threshold or exact P values were reported. In lines 219, 222, 265, 696, and 811, the reported values seem to clearly be thresholds (< a standard cutoff), while in lines 291,293, 297,300, values appear to be exact but are reported as thresholds (<).

This is not an error but rather reflects two different types of analysis. The predictions per locus (originally lines 219, 222 etc) are evaluated using an empirical P-value based on 1000 permutations. As such, they are thresholded at 1/1000. The overlap with open chromatin regions, on the other hand, are based on a z-score with the P-values taken from a standard conversion of z-scores to P-values.

Page 13/Table 2: At face value, it seems surprising that the overlap between Dmel SCRMshaw predictions with open chromatin is so much smaller than the overlap between predictions and open chromatin in other species, both in raw % (Tcas, D plexippus, H. himera) and fold enrichment (Tcas), given that the training sets for SCRMshaw are all derived from Dmel data. The discussion here does not touch on this aspect of the results, and the interpretation of this approach, in general, would be strengthened if the authors could comment on potential reasons why this pattern may be arising here, or at least acknowledge that this is an open question.

There are many variables at play here, as the data are from different species, from different tissues, and from different methods. Thus we think it is difficult to read too much into the precise results from these comparisons—the main take-home is really just that there is a significant amount of overlap. In acknowledgment of this, we have slightly modified the text in this section so that it now notes (line 302ff): “These comparisons are imperfect, as the tissues used to obtain the chromatin data do not precisely correspond to the training sequences used for SCRMshaw, and the data were obtained using a variety of methods.”

Line 318-329: The inferences from the reporter gene assay deserve a more nuanced treatment than they are given here. The important nuance that was not addressed by the discussion here is that the imaginal disc mode of development in *Drosophila* is not broadly representative of the development of larval/adult epithelial tissues across Holometabola; thus, inference of a true positive validation becomes complicated in cases where predicted enhancers from a species were tested and shown to drive expression in a fly imaginal disc that the native species have no direct disc counterpart to. For example, in line 388 a Tcas enhancer is reported to drive expression in the eye-antennal disc, and in lines 404 and 423 additional Tcas enhancers were reported to drive expression in the leg discs; however, Tribolium larvae do not possess antennal discs or leg discs set aside during embryogenesis in the sense that flies do - instead the homologous epithelial tissues form larval antennae and larval legs external to the body wall that are actively used at this life stage and are starkly different in morphology than an internally invaginated epithelial disc, that will directly give rise to adult tissues in subsequent molts. Is the interpretation of an expression pattern driven in a fly disc as a true positive really as straightforward as it was presented here, when in the native species the expression pattern driven by the enhancer in question would be in the context of an extremely different tissue morphology? That said, I understand and am deeply sympathetic to the constraints on the authors in performing transgenic experiments outside of the model fly; but these divergent modes of development across Holometabola deserve a mention and nuance in the interpretation here.

This is indeed a very important point, and we greatly appreciate Reviewer 3 pointing out this caveat when interpreting the outcomes of our cross-species reporter assay. Reviewer 3 is correct that the imaginal disc mode of adult tissue (i.e. imaginal) development found in Diptera does not represent the imaginal development across Holometabola.

In fact, imaginal development is quite diverse among Holometabola. For instance, larval leg and antennal cells appear to directly develop into the adult legs and antennae in Coleoptera (i.e. primordial imaginal cells function as larval appendage cells), while some cells within the larval legs and antennae are set aside during larval development specifically for adult appendages in Lepidopteran species (i.e. imaginal cells exist within the larval appendages but do not contribute to the formation of larval appendages). In contrast, an almost entire set of cells that develop into adult epithelia are set aside as imaginal discs during embryogenesis in Diptera. Furthermore, the imaginal disc mode of development appears to have evolved independently in

Hymenoptera. Therefore, determining how imaginal primordial tissues correspond to each other among Holometabola has been a challenging task and a topic of high interest within the evo-devo and entomology communities.

Nevertheless, despite these differences in mode of imaginal development, decades of evo-devo studies suggest that the gene regulatory networks (GRNs) operating in imaginal primordial tissues appear to be fairly well conserved among holometabolan species (for example, see Tomoyasu et al. 2009 regarding wing development and Angelini et al. 2012 regarding leg development between flies and beetles). These outcomes imply that a significant portion of the transcriptional landscape might be conserved across different modes of imaginal development. Therefore, an enhancer functioning in the Tribolium larval leg tissue (which also functions as adult leg primordium) could be active even in the leg imaginal disc of *Drosophila*, if the trans factors essential for the activation of the enhancer are conserved between the two imaginal tissues.

That being said, we fully expect there to be both false negative and false positive results in our cross-species reporter assay. We are optimistic about the biological relevance of the positive outcomes of our crossspecies reporter assay, especially when the enhancer activity recapitulates the expression of the corresponding gene in *Drosophila* (for example, Am_ex Fig6B and Tc_hth Fig7B). Nonetheless, the biological relevance of these enhancer activities needs to be further verified in the native species through reporter assays, enhancer knock-outs, or similar experiments.

In recognition of the Reviewer’s important point, we added the following caveat in our Discussion (lines 549553): “Furthermore, the unique imaginal disc mode of adult epithelial development in *D. melanogaster* might have prevented some enhancers of other species from working properly in *D. melanogaster* imaginal discs, likely producing additional false negative results. Evaluating enhancer activities in the native species will allow us to address the degree of false negatives produced by the cross-species setting.” We moreover mention this caveat in the Results section when we first introduce the reporter assays (line 342).

Line 580: This is the first time that the weakness of the closest-gene pairing approach is mentioned. This deserves mention earlier in the manuscript, as unfortunately, this is one of the major bottlenecks to this and any other approaches to investigating enhancer function. Could the authors address this earlier, perhaps pages 7-8, and provide citations for current understanding in the field of how often closest-gene pairing approaches correctly match enhancers to target genes?

We have added text as suggested on p.7-8 acknowledging the shortcomings of the closest-gene approach. We also clarify at the end of that section (lines 173-181) that target gene assignments, while useful for interpretation, have no bearing on the enhancer predictions themselves (which are generated prior to the target gene assignment steps).